# Glia-derived ATP inversely regulates excitability of pyramidal and CCK-positive neurons

Zhibing Tan[1,2,*], Yu Liu[1,*], Wang Xi[1], Hui-fang Lou[1], Liya Zhu[1], Zhifei Guo[1], Lin Mei[3] & Shumin Duan[1]

Astrocyte responds to neuronal activity with calcium waves and modulates synaptic transmission through the release of gliotransmitters. However, little is known about the direct effect of gliotransmitters on the excitability of neuronal networks beyond synapses. Here we show that selective stimulation of astrocytes expressing channelrhodopsin-2 in the CA1 area specifically increases the firing frequency of CCK-positive but not parvalbumin-positive interneurons and decreases the firing rate of pyramidal neurons, phenomena mimicked by exogenously applied ATP. Further evidences indicate that ATP-induced increase and decrease of excitability are caused, respectively, by P2Y1 receptor-mediated inhibition of a two-pore domain potassium channel and A1 receptor-mediated opening of a G-protein-coupled inwardly rectifying potassium channel. Moreover, the activation of ChR2-expressing astrocytes reduces the power of kainate-induced hippocampal *ex vivo* gamma oscillation. Thus, through distinct receptor subtypes coupled with different $K^+$ channels, astrocyte-derived ATP differentially modulates the excitability of different types of neurons and efficiently controls the activity of neuronal network.

[1] Department of Neurobiology, Key Laboratory of Medical Neurobiology of The Ministry of Health of China, Key Laboratory of Neurobiology of Zhejiang Province, Zhejiang University School of Medicine, Hangzhou 310058, China. [2] Institute of Neuroscience and Key Laboratory of Neuroscience, Shanghai Institutes for Biological Sciences, Chinese Academy of Sciences, Shanghai 200031, China. [3] Department of Neuroscience and Regenerative Medicine, Medical College of Georgia, Augusta University, Augusta, Georgia 30912, USA. * These authors contributed equally to this work. Correspondence and requests for materials should be addressed to S.D. (email: duanshumin@zjuem.zju.edu.cn).

Astrocytes, the most abundant cell type in the brain, have important roles in the central nervous system, including synaptogenesis, neuronal metabolism and regulating the homeostasis of extracellular ions and neurotransmitters, as well as modulating synaptic transmission and plasticity[1–4]. The processes of astrocytes enwrap synapses to form a structure known as the tripartite synapse[5–7], where they respond to synaptic activity with increasing intracellular $Ca^{2+}$ and, in turn, regulate neuronal activity by releasing various gliotransmitters. ATP is one of the major diffusible signalling molecules released by astrocytes[8,9]. Previous studies have shown that astrocyte-derived ATP, together with its degradation product adenosine, regulates synaptic transmission through a presynaptic mechanism[10–13]. Apart from synapses, accumulating evidence also shows that ATP modulates neuronal excitability[14–18]. However, the effects of endogenous ATP on the activity of the intact neural network and the underlying mechanisms have not been fully characterized.

Purinoceptors are broadly divided into adenosine (P1) and ATP (P2) receptors. P1 receptors are G-protein-coupled and classified into four subtypes: A1 and A3 receptors are generally coupled to $G_{i/o}$, whereas A2A and A2B are linked to $G_s$ (refs 9,19). P2 receptors are divided into ionotropic P2X and metabotropic P2Y receptors. Eight subtypes of P2Y receptors have been cloned in mammals. $P2Y_{1,2,4,6,11}$ activate phospholipase C via $G_{q/11}$, while the others stimulate or inhibit adenylyl cyclase via $G_s$ ($P2Y_{14}$) or $G_{i/o}$ ($P2Y_{12,13}$). Multiple subtypes of purine receptors have been found throughout the hippocampus[19], but their integrative functions in modulating neural network activity are not well studied.

Controlling the opening and closing of $K^+$ channels is a strategy used by a wide range of factors, including G-protein-coupled receptors, to modulate neuronal activity and signal propagation throughout the nervous system[20–22]. Exogenous ATP has been shown to modulate the activity of the M-channel (KCNQ)[23], $Ca^{2+}$-activated $K^+$ channel ($K_{Ca}$; ref. 24), G-protein-coupled inwardly-rectifying $K^+$ channel (GIRK)[21], and two-pore domain $K^+$ channel (K2P; ref. 22). Despite this, most of these results were obtained from heterologous expression studies and their physiological and pathological relevance remains to be explored.

A major challenge for studying the specific roles of astrocytes is the lack of efficient approaches to selectively stimulate them in the brain. To achieve this, we specifically expressed the light-gated $Ca^{2+}$-permeable channel channelrhodopsin-2 (ChR2; refs 10,25,26) in astrocytes. We find that selective stimulation of astrocytes via ChR2 results in increased excitability of cholecystokinin (CCK) interneurons mediated by closing of K2P through the activation of P2Y1 receptors. In contrast, the same stimulation decreases the excitability of pyramidal neurons due to opening of GIRK through the activation of A1 receptors.

## Results

### Light activation of astrocytes changes neuronal excitability.
We took advantage of GFAP-cre mice to specifically express ChR2-mCherry in astrocytes in the hippocampal CA1 area. Anti-RFP antibody was used to highlight the area of ChR2 expression (Supplementary Fig. 1a,b). Immunostaining showed that ChR2-mCherry co-localized with the astrocyte-specific marker GFAP, but not with the neuronal marker MAP2 and the NG2-glial marker NG2 (Supplementary Fig. 1c,d). The cells expressing ChR2-mCherry exhibited passive membrane properties typical of astrocytes[27] and were reliably activated by blue light (Supplementary Fig. 1e,f).

Interneurons and pyramidal neurons in the CA1 area were identified based on their location, shape and firing properties.

The firing rate of action potentials (APs) was taken as an indication of neuronal excitability[15,18]. Depolarizing currents (50-100 pA) were injected into neurons to maintain AP firing at 0.5–1.5 Hz. Neuronal excitability was monitored before, during and after blue light stimulation (500 ms pulses at 1 Hz for 2 min). To exclude the influence of synaptic transmission, 0.5 mM kynurenic acid (an ionotropic glutamate receptor antagonist) and 10 μM bicuculline (a $GABA_A$ receptor antagonist) were applied to all experiments unless where specified. We found that the firing rate of a subpopulation of interneurons (12 of 25) increased after light stimulation, while the firing frequency decreased in all pyramidal neurons recorded. In a parallel experiment, the same intensity of blue light was used to stimulate astrocytes expressing EGFP (enhanced green fluorescent protein), and no significant AP frequency change occurred in interneurons and pyramidal neurons (Fig. 1a–c). Neuronal excitability was also assayed by a series of 2-s step-current injections and the AP numbers at each step were compared. The results showed that the AP number increased in interneurons (6 of 11) and decreased in pyramidal neurons recorded immediately after 2-min illumination (Fig. 1d,e, and Supplementary Fig. 2).

Astrocytes release glutamate, D-serine, ATP and other factors in response to various stimuli[28–30]. The phenomena observed here is unlikely to be mediated by glutamate, because an ionotropic glutamate receptor antagonist was present in the perfusate and the broad-spectrum metabotropic glutamate receptor (mGluR) antagonist AP-3 showed no effect (detailed later). Astrocyte-derived D-serine has a critical role in long-term potentiation but not neuronal excitability by acting as a co-agonist for N-methyl-D-aspartate (NMDA) receptors[29,31]. Thus, we focused on the effects of astrocyte-derived ATP. Using a ATP-specific biosensor[15], we detected that light stimuli increased extracellular ATP concentration up to approximately 1 μM ($0.93 \pm 0.13$ μM) in ChR2-expressing slices, but had no effect in EGFP-expressing slices (Fig. 1f and Supplementary Fig. 3). This is consistent with previous studies, which showed that activation of astrocytes increases the extracellular ATP concentration up to 1 μM (refs 15,32). We therefore directly applied 1 μM ATP to neurons by gravity through a glass micropipette (tip: 50 μm). Similar to the results obtained in light-stimulated astrocytes, we found that a subgroup of interneurons (6 of 13) was excited and all the pyramidal neurons were inhibited by exogenously applied ATP (Fig. 1g–i). The peak effects of the responses of exogenous ATP ($1.78 \pm 0.22$ min for interneurons, $1.57 \pm 0.13$ min for pyramidal neurons) is much faster than light stimulation ($3.5 \pm 0.2$ min for interneurons, $2.86 \pm 0.17$ min for pyramidal neurons; Fig. 1j), possibly reflecting the delayed calcium elevation and propagation following light stimulation (Fig. 2a–g, Supplementary Movies 1 and 2), as well as a diffusion delay of the light-induced ATP released from astrocytes.

To facilitate analysis of ion channels underlying ATP-induced responses, we puff-applied 100 μM ATP by a picospritzer through a pipette with a tip $\sim 8$ μm in diameter. In this condition, a fast and robust depolarization ($9.26 \pm 0.32$ mV, 8 of 15) together with an eruption of AP firing occurred in a subgroup of interneurons (Supplementary Fig. 4a). In contrast, hyperpolarization was found in all the pyramidal neurons recorded ($-4.46 \pm 0.23$ mV, 10 of 10). Meanwhile, a transient decrease followed by a sustained increase in the frequency and amplitude of spontaneous postsynaptic potentials (sPSPs) also occurred in pyramidal neurons (Supplementary Fig. 4b–f). The decreased sPSPs may reflect the direct inhibitory effect of ATP on synaptic transmission through presynaptic A1 receptors[33]. The increased sPSPs were blocked by 10 μM bicuculline, suggesting that most of them were inhibitory (Supplementary Fig. 4c). Furthermore, 0.5 μM tetrodotoxin,

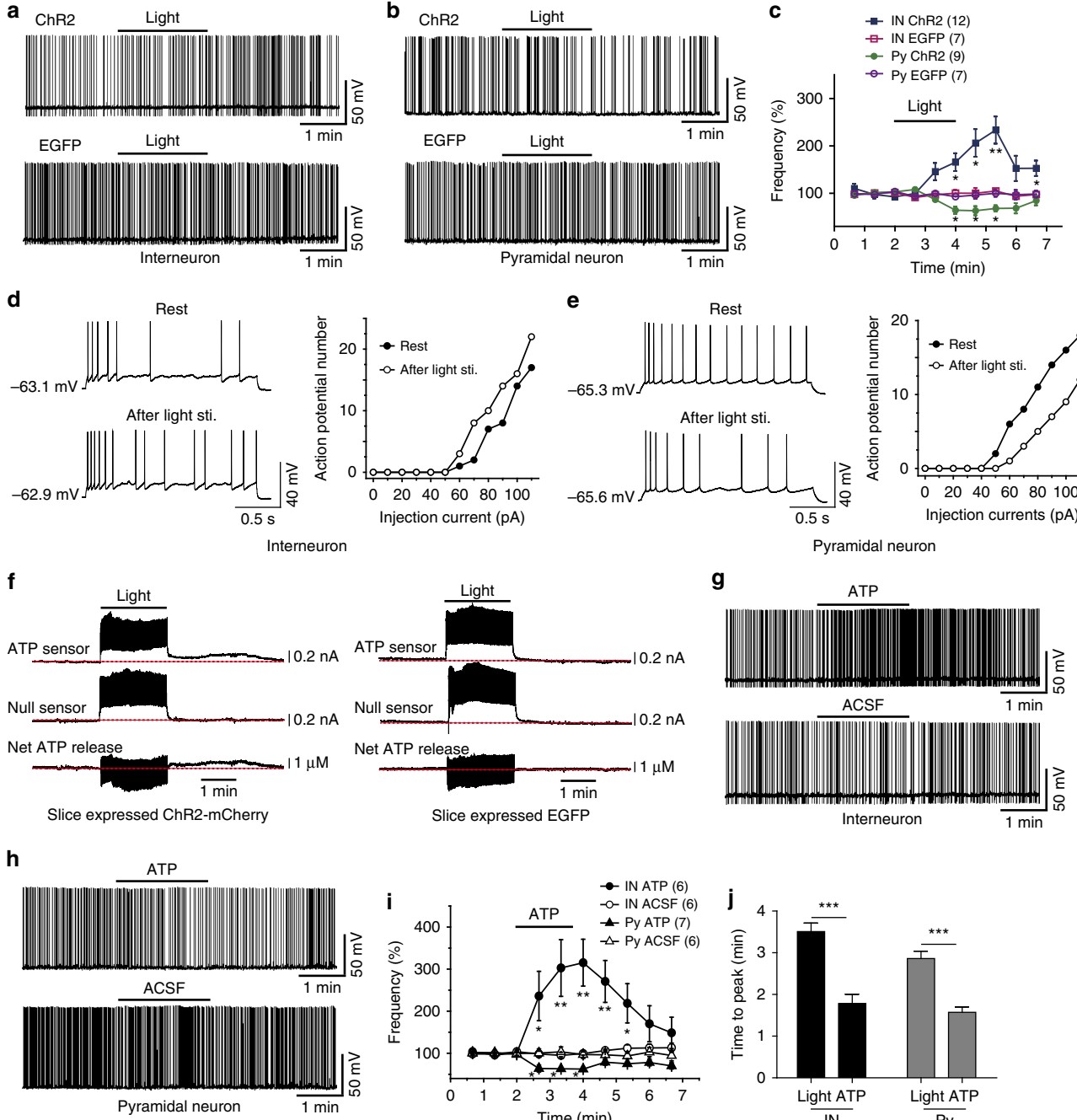

**Figure 1 | Light stimulation of ChR2-expressing astrocytes modulates neuronal excitability by releasing ATP.** (**a,b**) Sample traces showing AP firing of interneurons and pyramidal neurons before, during and after 2 min of light stimulation (1 Hz, 0.5 s on and 0.5 s off) of astrocytes expressing ChR2 (upper) or EGFP (lower). (**c**) Summary data showing the time courses of AP frequency of interneurons (IN) and pyramidal neurons (Py) before, during and after light stimulation (bar) of slices expressing ChR2 or EGFP in astrocytes (two-way analysis of variance (ANOVA), for interneuron $F_{(9,177)} = 2.98$, $P = 0.0025$; for pyramidal neuron $F_{(9,130)} = 3.1$, $P = 0.0021$). Data are normalized to the AP frequency averaged from 2 min recording before light stimulation for each recording. (**d,e**) Left: representative APs of an interneuron (**d**) and a pyramidal neuron (**e**) in response to current injection before (upper) and after (lower) 2 min light stimulation (sti.). Right: plots of AP numbers from one typical neuron in response to a serious current injection before (filled symbols) and after (open symbols) light stimulation. (**f**) Represent traces recorded by ATP-specific biosensors. Net current evoked by light-induced ATP release was calculated by subtracting null sensor-recorded current from ATP sensor-recorded current. The red dotted lines at the bottom traces depict the basal ATP level before light stimulation. Of note, the spike-like noise that appeared during light stimuli was artefact elicited directly by light. (**g,h**) Sample traces showing AP firing in interneuron and pyramidal neuron before, during and after puffing 1 μM ATP (upper) or HEPES-buffered aCSF (lower) for 2 min. (**i**) Summary data showing the time courses of AP frequency of interneurons (IN) and pyramidal neurons (Py) before, during and after puffing 1 μM ATP (filled symbols) or HEPES buffered aCSF (open symbols) for 2 min (bar) (two-way ANOVA, for interneuron $F_{(9,100)} = 2.21$, $P = 0.0274$; for pyramidal neuron $F_{(9,105)} = 2.33$, $P = 0.021$). (**j**) Time to peak of responses induced by light stimuli and ATP application estimated by period from the time point starting to apply ATP or light to the time inducing its largest effect (Student's t-test, IN: $P < 0.0001$; Py: $P < 0.0001$). Error bars are defined as s.e.m. in all the figures. *$P < 0.05$, **$P < 0.01$, ***$P < 0.001$

which blocks AP firing, also abolished the increased sPSPs (Supplementary Fig. 4d), supporting the notion that the increased sPSPs were mediated by enhanced interneuron excitability instead of direct modulation of synaptic transmission.

**Light stimuli elevate intracellular $Ca^{2+}$ level in astrocytes.** Astrocytes release gliotransmitters in a $Ca^{2+}$-dependent manner. We thus took the advantage of calcium imaging to investigate

whether light stimuli change calcium level in astrocytes. After bulk loading calcium dyes in astrocytes, we found that light stimuli indeed increased the number of calcium waves in slices expressing ChR2 but not in slice expressing EGFP (Fig. 2a–c,g, Supplementary Fig. 5, Supplementary Movies 1 and 2). This effect sustained for several minutes after stimuli were terminated, a time course consistent with neuronal excitability changes. To further determine whether the neuronal responses were astrocytic $Ca^{2+}$-dependent, we loaded BAPTA into astrocytes located near the

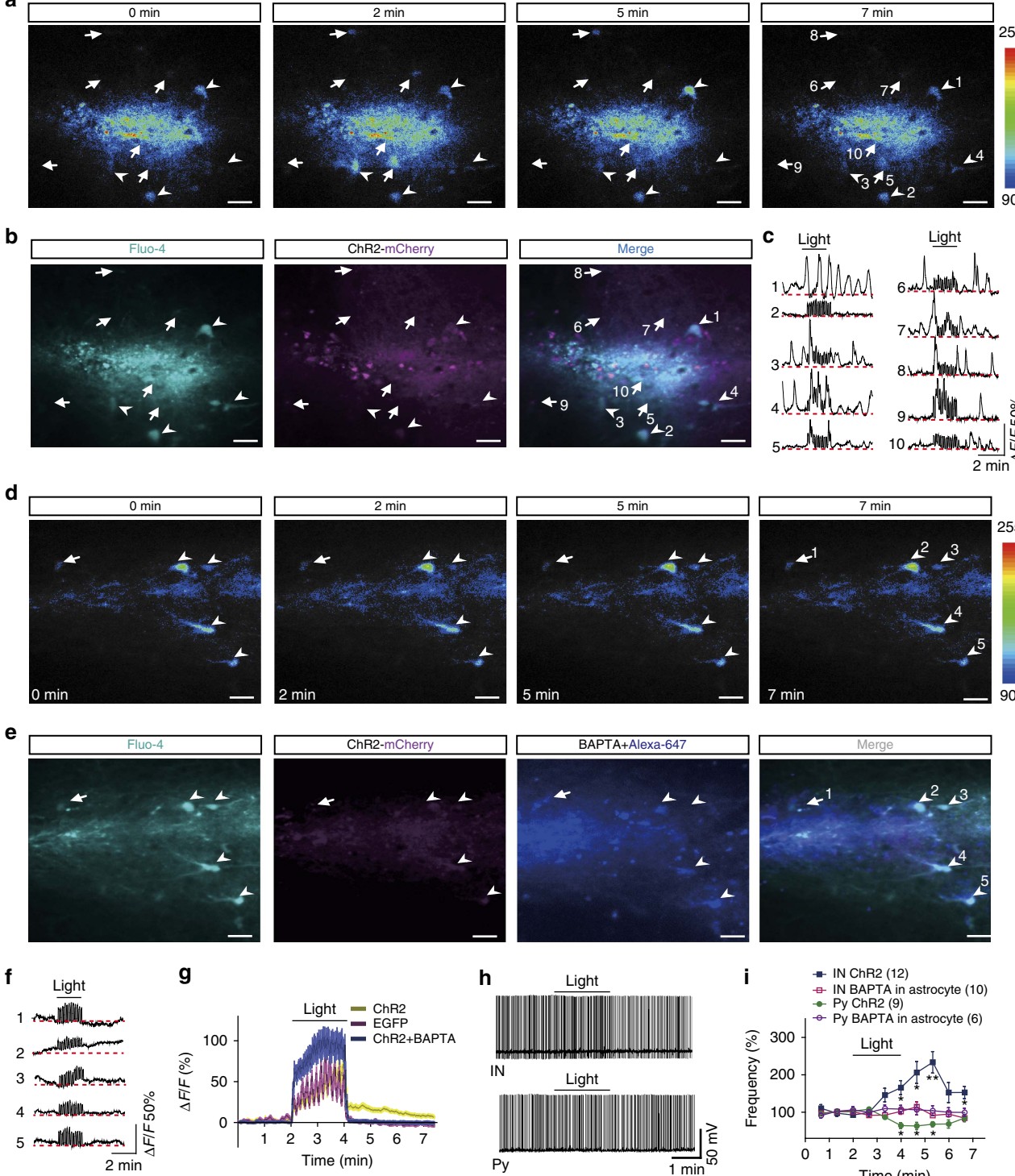

ChR2-mCherry-expressing area in the stratum radiatum. This was achieved by repetitive injections of small depolarizing currents (10 pA, 100 ms, 2 Hz) through a whole-cell recording pipette loaded with BAPTA (15 mM) together with Alexa Fluor-647 (100 μM), a blue fluorescent dye with a molecular weight and charge similar to BAPTA. This method allows the rapid spread of Alexa Fluor-647 and BAPTA (within 10 min) across many astrocytes via gap junctions. After this astrocytic loading with BAPTA, we found that the astrocyte calcium waves and the neuronal excitability changes induced by light stimulation of astrocytes were completely blocked (Fig. 2d–i, Supplementary Movie 3).

**ATP mainly excites cholecystokinin-positive interneurons.** Interneurons in the hippocampus can be divided into several subgroups based on $Ca^{2+}$-binding proteins and neuropeptide expression[34]. We found that only ~50% (269 of 540) of interneurons in the CA1 stratum radiatum (SR) and stratum lacunosum-moleculare (SLM) were excited either by astrocyte stimulation or by direct application of exogenous ATP, indicating that ATP might excite a specific type of interneuron. Although only half of the regularly spiking interneurons responded to ATP, all of the ATP-excited interneurons were regularly spiking neurons, none of them was fast-spiking parvalbumin (PV)-positive interneurons (0 of 19; Fig. 3a).

We then used single-cell reverse transcription polymerase chain reaction (RT–PCR) and studied the expression of the interneuron markers calbindin (CB), calretinin (CR), cholecystokinin (CCK), vasoactive intestinal polypeptide (VIP) and neuropeptide Y (NPY) in recorded interneurons in an independent experiment (Supplementary Fig. 6). Neurons with depolarization >2 mV were referred to as positive, whereas others were regarded as negative. The results showed that among all the interneurons excited by ATP, 83.3% (25 of 30) expressed CCK, 40% (12 of 30) expressed CB, 13.3% (4 of 30) expressed CR, 16.7% (5 of 30) expressed VIP and 23.3% (7 of 30) expressed NPY (Fig. 3b,c). There are overlaps among these five interneuron markers in the CA1 area[34,35]. We found that among ATP-excited interneurons, 83.3% of the CB-positive interneurons (10 of 12), 50% of the CR-positive interneurons (two of four), 80% of the VIP-positive interneurons (four of five) and 85.7% of the NPY-positive interneurons (six of seven) also co-expressed CCK (Fig. 3c). These results suggest that most of the ATP-excited interneurons are CCK-positive, while 22.4% (11 of 49) of the CCK-positive interneurons does not respond to ATP and 19.4% (7 of 36) of the ATP-excited interneurons does not express CCK (Fig. 3d). This might have resulted from experimental errors that occasionally occur in single-cell RT–PCR experiments[36]. The involvement of CCK-positive interneurons was further verified by immunostaining. Recorded interneurons were labelled with 1% biocytin and CCK-specific antibody staining showed that 91.7% (11 of 12) of the ATP-excited interneurons were CCK-positive (Fig. 3e,f). These results demonstrate that most, if not all, of the ATP-responsive interneurons express CCK.

**ATP effects are through P2Y1 and A1 receptors.** ATP receptors include ionotropic P2X receptors and G-protein-coupled P2Y receptors. Several subtypes of these receptors are expressed throughout the hippocampus[19]. We found that puffing the P2Y-specific agonist 2-(methylthio) adenosine 5′-diphosphate (MeS-ADP, 100 μM) into an interneuron induced a robust depolarization comparable to puffing 100 μM ATP, while puffing the P2X-specific agonist α,β-methyleneadenosine 5′-triphosphate (AMP-CPP, 100 μM) had no detectable effect on the same neuron (Fig. 4a). Moreover, the P2Y receptor antagonists suramin (100 μM) and pyridoxal phosphate-6-azo benzene-2,4-disulfonic acid (PPADS, 50 μM), but not the mGluR antagonist AP-3, reversibly inhibited the ATP-induced depolarization (Fig. 4b). Consistent with the previous report, these experiments indicate that P2Y, not P2X receptors, are responsible for the ATP-induced excitation of interneurons[14]. Furthermore, the P2Y1-specific antagonist N6-methyl-2′-deoxyadenosine-3′, 5′-bisphosphate (MRS2179, 20 μM) also reversibly inhibited the ATP-induced depolarization (Fig. 4b). No interneurons were depolarized by ATP in P2Y1 knockout mice (Fig. 4c), strongly supporting the conclusion that ATP-induced depolarization is mediated by activation of P2Y1 receptors. Consistent with this notion, light-induced increase of interneuron firing was also abolished in the presence of suramin, PPADS or MRS2179 (Fig. 4e,f). Moreover, light stimulation of astrocytes expressing ChR2 did not increase the interneuron firing in P2Y1 knockout mice (Fig. 4h,i).

In contrast to depolarization, the ATP-induced hyperpolarization in pyramidal neurons was resistant to the P2 receptor antagonists reactive blue 2 (RB-2, 100 μM), PPADS (50 μM) and MRS2179 (20 μM). $P2Y_{12}$ and $P2Y_{13}$ are $G_{i/o}$-coupled receptors linked to GIRK and are expressed in the hippocampus[19]. Interestingly, $P2Y_{12}$-specific antagonist AR-C 66096 (30 μM) and the $P2Y_{13}$-specific antagonist MRS2211 (100 μM) did not inhibit the ATP-induced hyperpolarization (Supplementary Fig. 4g). ATP-induced excitation of interneurons may release abundant GABA, which could secondarily hyperpolarize pyramidal neurons. Since ATP induced hyperpolarization in the presence of bicuculline, we investigated whether $GABA_B$

**Figure 2 | Light stimuli induced prolonged calcium increase in astrocytes that can be blocked by astrocytic loading of BAPTA.** (**a**) Example time-lapse images of $Ca^{2+}$ signals. Light stimuli were applied at time point 2 min and lasted for 2 min. (**b**) Confocal images showing the ChR2-expressing astrocytes loaded with the $Ca^{2+}$ fluorescent dye Fluo-4. Images were sampled from the same field as in **a**. Scale bars in **a** and **b** indicate 20 μm. (**c**) Example traces of light-induced $Ca^{2+}$ signals (ΔF/F) in astrocyte as numbered in **a** and **b**. Note that light stimulation initially induced $Ca^{2+}$ elevation in ChR2-expressing astrocytes (traces 1, 3, 4), which then propagated to non-ChR2-expressing cells (traces 5, 7, 8, 9). The noise appearing during light stimulation reflect sampled artificial signals caused by blue light stimulation. Dotted red lines indicate base level of $Ca^{2+}$ signals. (**d**) Example time-lapse images of $Ca^{2+}$ signals in astrocytes loaded with BAPTA. (**e**) Confocal images showing the ChR2-expressing astrocytes bulk loaded with the $Ca^{2+}$ fluorescent dye Fluo-4 and intracellularly loaded BAPTA together with Alexa 647. Images were sampled from the same field as in **d**. (**f**) Example traces of light-induced $Ca^{2+}$ signals (ΔF/F) in astrocyte as numbered in **d** and **e**. Dotted red lines indicate base level of $Ca^{2+}$ signals. (**g**) Summary data showing the changes in calcium signals (ΔF/F) in slices expressing ChR2, EGFP or ChR2 loaded with BAPTA. Data are averaged from more than six slices in three independent experiments for each condition. Note elevated calcium level after light stimulation in ChR2 group (yellow trace). (**h**) Sample traces showing AP firing of interneuron (upper) and pyramidal neuron (lower) before, during and after 2 min of light stimulation (1 Hz, 0.5 s on and 0.5 s off) of astrocytes expressing ChR2 and loaded with BAPTA. (**i**) Summary data showing the time courses of AP frequency of interneurons (IN) and pyramidal neurons (Py) before, during and after light stimulation (bar) of slices expressing ChR2 with and without BAPTA loading in astrocytes (two-way analysis of variance, for interneuron $F_{(9,207)} = 4.89$, $P < 0.0001$; for pyramidal neuron $F_{(9,110)} = 2.26$, $P = 0.023$).

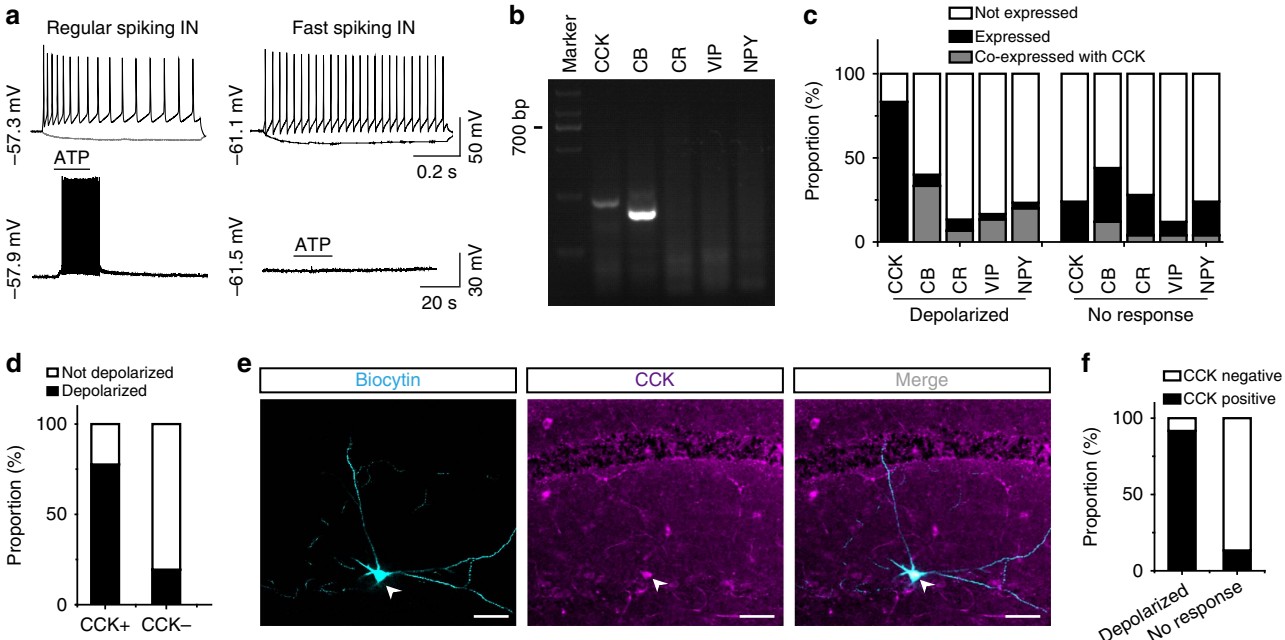

**Figure 3 | ATP-excited neurons are mainly CCK-positive interneurons.** (**a**) Sample traces showing the AP discharge pattern of an ATP-responsive CCK interneuron (left) and ATP non-responsive fast-spiking interneuron (right). APs in upper traces were generated by injection of 100 pA depolarizing current. (**b**) Representative agarose gel electrophoresis results showing the expression of the interneuron markers CCK, CB, CR, VIP and NPY in an ATP-responsive interneuron. The final PCR product of CCK is 234 bp. (**c**) Summary data showing the proportion of interneuron marker expression in neurons responsive or unresponsive to ATP. Black, expression of indicated markers; white, no markers; grey, markers co-expressed with CCK. (**d**) Summary data showing the proportion of CCK-containing interneurons responsive to ATP. 'CCK + ', neurons expressing CCK; 'CCK − ', neurons without expression. Black, neurons depolarized by ATP; white, neurons not affected. (**e**) Confocal images showing that a biocytin-filled interneuron (arrowhead) responding to ATP was CCK-positive. Scale bars, 50 μm. (**f**) Summary of immunostaining results showing the proportion of ATP-excited interneurons that were CCK-positive. Black, CCK expression; white, no CCK expression.

receptors were involved and found that the GABA$_B$ receptor-specific antagonist phaclofen (100 μM) did not inhibit the ATP-induced hyperpolarization (Supplementary Fig. 4g).

ATP has been reported to be rapidly degraded extracellularly by ectonucleotidases into adenosine, which in turn inhibits glutamatergic synaptic transmission[11,13]. Consistent with these findings, we found that the A1 receptor-specific antagonists 1,3-dipropyl-8-cyclopentylxanthine (DPCPX) and 8-cyclopentyl-1,3-dimethylxanthine (CPT) at 1 μM irreversibly inhibited the ATP-induced hyperpolarization (Fig. 4d). Furthermore, the decrease of pyramidal neuron firing induced by light stimulation of astrocytes expressing ChR2 was also inhibited by 1 μM DPCPX and CPT (Fig. 4e,g). Thus the hyperpolarizing effect of ATP might be mediated by adenosine owing to the high expression of ectonucleotidases in the hippocampus[9].

**ATP excites CCK interneurons by inhibiting K2P channels.** To identify the ion channel responsible for the ATP-induced depolarization in interneurons, we used a ramp protocol (from − 130 mV to − 60 mV, at 10 mV s$^{-1}$) to construct the voltage–current curve of the ATP-induced current[37]. The extracellular solution was supplemented with (in μM) 10 DNQX, 50 APV, 10 bicuculline and 100 Cd$^{2+}$ plus 0.5 μM tetrodotoxin to block synaptic transmission and some voltage-gated Ca$^{2+}$ and Na$^+$ channels. The reversal potential of the ATP-induced inward current was − 83.3 ± 2.4 mV, which is close to the calculated K$^+$ reversal potential (− 89.4 mV) in our experimental conditions where the extracellular K$^+$ concentration was 3.5 mM (Fig. 5b), suggesting that ATP excites interneurons through the inhibition of background K$^+$ channels.

This hypothesis was further supported by the finding that the input resistance of interneurons increased during light stimuli and ATP application (Fig. 5c,d, Supplementary Fig. 7a,b). The ATP-induced depolarization was not apparently affected by the nonselective cation channel inhibitors Cd$^{2+}$ and La$^{3+}$ (0.5 mM), by substitution of extracellular Na$^+$ with the same concentration of choline, or by omission of extracellular Ca$^{2+}$ (Fig. 5a). These results exclude the involvement of nonselective cation channels, Na$^+$ channels or Ca$^{2+}$ channels.

Next, we sought to determine what type of K$^+$ channel is accounted for the ATP-induced interneuron depolarization. We found that none of the following conventional K$^+$ channel-blockers inhibited the ATP effect: the voltage-gated K$^+$ channel blocker 4-AP, the ATP-sensitive K$^+$ channel blocker tolbutamide, the K$_{Ca}$ blocker apamin and the Na$^+$–K$^+$–Cl$^-$ co-transporter blocker bumetanide (Supplementary Fig. 8c). Interestingly, acetylcholine (ACh) occluded the ATP-induced depolarization in a concentration-dependent manner in which 0.5 mM had the maximal effect (Fig. 5e). Furthermore, the metabotropic-type (M-type) ACh receptor agonist muscarine (50 μM) reversibly blocked the ATP effect (Fig. 5e), suggesting that ACh occluded the ATP effect by activating M-type ACh receptors. ACh has been reported to modulate the function of several kinds of K$^+$ channels, of which the most studied one is the M-current mediated by KCNQ expressed in cerebellar granule cells and the hippocampus[38,39]. We recorded M-currents using a typical protocol for detecting standing outward K$^+$ currents ($I_{kso}$). Although ACh and muscarine occlude the ATP-induced current (Supplementary Fig. 8a,b), it was not an M-current, because the specific inhibitor XE991 blocked neither the ATP-sensitive current nor the ATP-induced depolarization

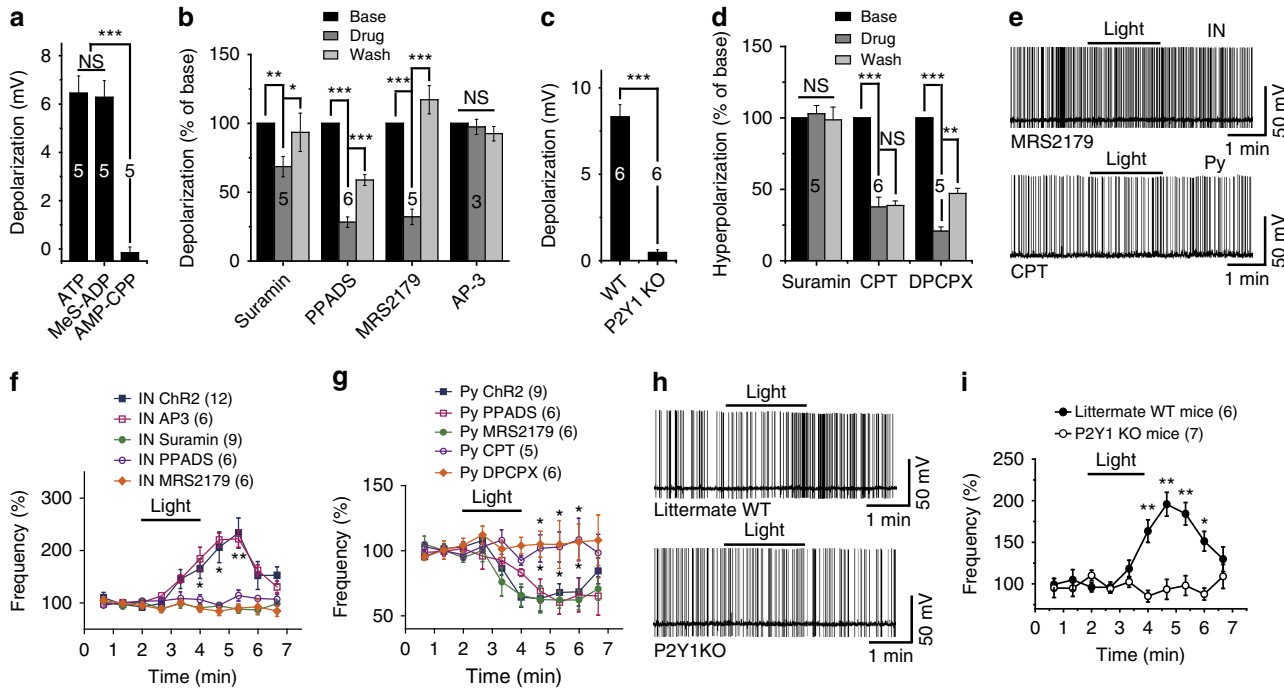

**Figure 4 | Receptors involved in the modulation of neuronal excitability by ATP.** (**a**) Summary of membrane potential changes after puffing with 100 μM ATP, 100 μM Mes-ADP or 100 μM AMP-CPP. Note that the three agonists were applied to the same interneurons. Student's $t$-test, not significant (NS) $P = 0.434$, ***$P < 0.0001$. (**b**) Summary of normalized data showing the percentage inhibition of ATP-induced depolarization by P2 receptor and mGluR antagonists (Student's $t$-test, suramin: **$P = 0.007$, *$P = 0.035$, PPADS: ***$P < 0.0001$, ***$P < 0.0001$, MRS2179: ***$P = 0.0003$, ***$P = 0.0001$, AP-3: $P = 0.253$). (**c**) Summary of ATP-induced membrane potential change in interneurons in wild-type (WT) and P2Y1 knockout (KO) mice. The six most depolarized cells (6 of 13) were chosen for statistical analysis (Student's $t$-test, ***$P < 0.0001$). (**d**) Summary of normalized data showing the percentage inhibition of ATP-induced hyperpolarization by various antagonists (Student's $t$-test, suramin: $P = 0.46$, CPT: ***$P < 0.0001$, NS $P = 0.373$, DPCPX: ***$P < 0.0001$, **$P = 0.0057$). (**e**) Sample trace of interneuron and pyramidal neuron AP firing before, during, and after 2 min of light stimulation of astrocytes expressing ChR2 in the presence of 20 μM MRS2179 (upper panel) or 1 μM CPT (lower panel). (**f**) Summary data showing the time course of AP firing frequency of interneurons before, during and after light stimulation (bar) in the presence of AP-3 (two-way analysis of variance (ANOVA), $F_{(9,167)} = 0.2$, $P = 0.9938$), suramin (two-way ANOVA, $F_{(9,197)} = 4.56$, $P < 0.0001$), PPADS (two-way ANOVA, $F_{(9,167)} = 2.5$, $P = 0.011$), or MRS2179 (two-way ANOVA, $F_{(9,167)} = 3.16$, $P = 0.0015$). (**g**) Summary data showing the time course of pyramidal neuron AP firing frequency before, during and after light stimulation (bar) in the presence of PPADS (two-way ANOVA, $F_{(9,120)} = 0.94$, $P = 0.491$), MRS2179 (two-way ANOVA, $F_{(9,120)} = 0.41$, $P = 0.927$), CPT (two-way ANOVA, $F_{(9,110)} = 2.2$, $P = 0.0272$) or DPCPX (two-way ANOVA, $F_{(9,120)} = 2.11$, $P = 0.0336$). (**h**) Sample traces of interneuron AP firing before, during and after 2 min of light stimulation of astrocytes expressing ChR2 in littermate WT (upper) and P2Y1 KO mice (lower). (**i**) Summary data showing the time course of AP firing frequency of interneurons before, during and after light stimulation (bar) in littermate WT and P2Y1 KO mice (two-way ANOVA, $F_{(9,110)} = 7.42$, $P < 0.0001$).

(Fig. 5e and Supplementary Fig. 8b). Therefore, M-current inhibition may not be involved in the ATP-evoked increase in interneuron excitability.

ACh has also been shown to inhibit a newly defined two-pore domain K+ channel (K2P; ref. 40). Most of the subtypes of K2P channels are resistant to conventional K+ channel blockers but sensitive to quinidine and local anaesthetics[40,41]. We found that bath application of quinidine (100 μM), lidocaine (1 mM) or bupivacaine (1 mM) partially blocked the ATP-induced depolarization (Fig. 5e), suggesting that ATP excites interneurons by inhibiting K2P channels. Furthermore, in the presence of relatively low concentration (100 μM) of quinidine or lidocaine, the light-induced increase in the firing frequency of interneurons was also blocked (Fig. 5k,l), consistent with the involvement of K2P channels. In these experiments, 5–20 pA more current was injected into the recorded neurons to overcome the direct inhibition of quinidine and lidocaine on the action potential firing so that the effect of light stimulation could be tested. So far, 15 members of the K2P channel family have been identified in mammals. These channels are divided into six subfamilies based on their structural and functional properties, TWIK (TWIK1, TWIK2 and TWIK7), TASK (TASK1, TASK3 and TASK5),

TREK (TREK1, TREK2 and TRAAK), TALK (TALK1, TALK2 and TASK2), THIK (THIK1 and THIK2) and TRESK[40,42]. We then examined which subtypes of K2P are involved in the ATP-induced depolarization in interneurons. Because no specific blockers are available for each subtype of K2P, we studied the pharmacological profiles of the ATP-induced depolarization. We found that intracellular application of arachidonic acid (10 μM) partially blocked the ATP-induced depolarization (Fig. 5f). In addition, extracellular acidification (pH 6.4), but not intracellular acidification (pH 6.0), inhibited the ATP-induced depolarization (Fig. 5e and Supplementary Fig. 8d). Furthermore, both extracellular alkalization (pH 8.5) and the volatile anaesthetic isoflurane (100 μM) partially potentiated ATP-induced depolarization (Fig. 5g,h). Taken together, these results suggest that TASK subfamily of K2P channels may be involved[37]. We then examined the expression of the TASK family in the hippocampus. Since TASK-1 is mainly expressed in astrocytes[43] and TASK-5 alone does not form functional channels[44], we focused our efforts on TASK-3. We found that both hippocampal interneurons and pyramidal neurons expressed TASK-3, as revealed by single-cell RT–PCR and immunostaining (detailed later). However, the expression level of TASK-3 in

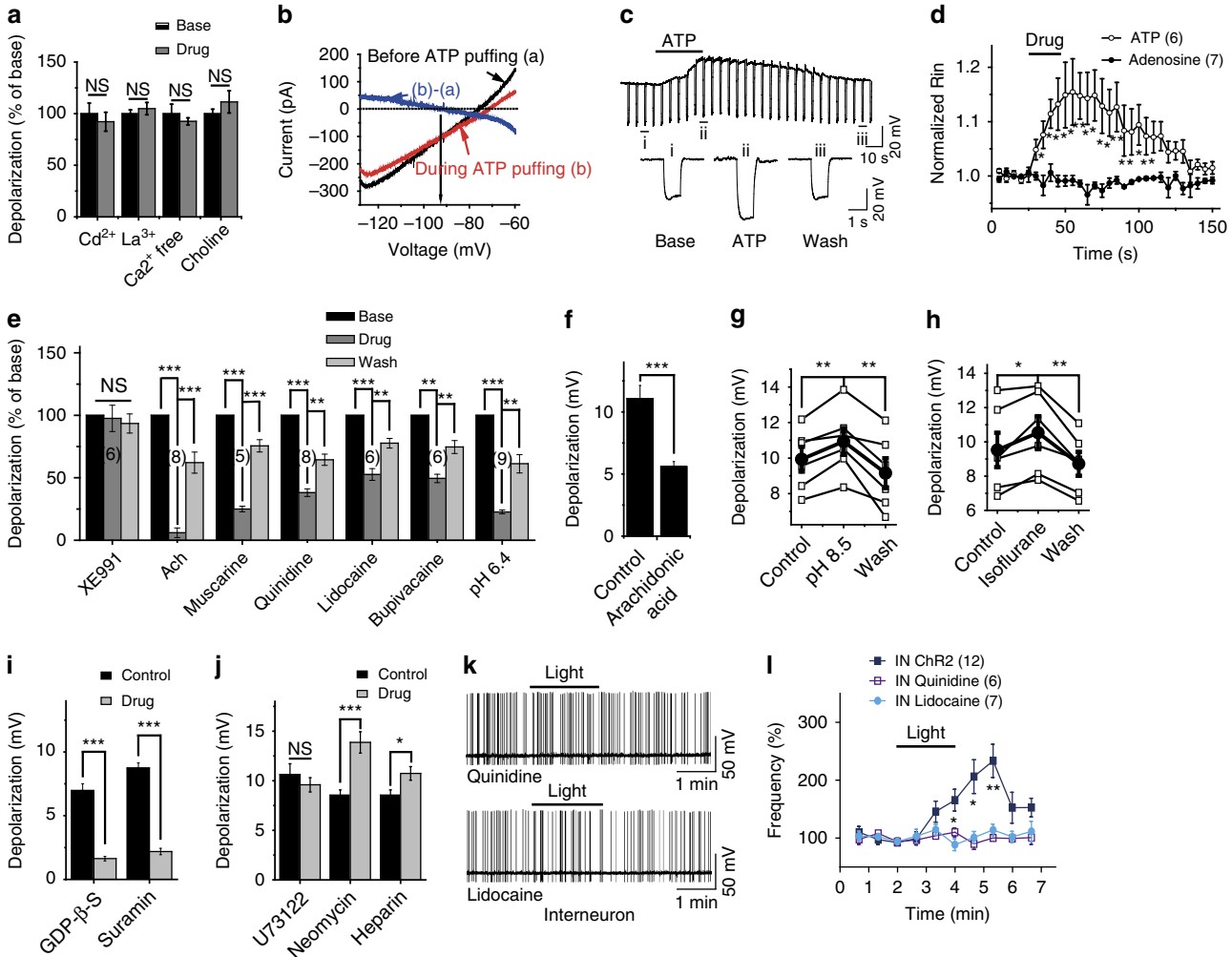

**Figure 5 | ATP excites CCK-positive interneurons by inhibiting K2P K$^+$ channels.** (**a**) Summary data showing the amplitude of ATP-induced depolarization in the presence of the nonspecific cation channel blockers Cd$^{2+}$ (Student's $t$-test, $P = 0.744$) and La$^{3+}$ (Student's $t$-test, $P = 0.193$), substitution of extracellular Na$^+$ with the same concentration of choline (Student's $t$-test, $P = 0.651$) or in the absence of extracellular Ca$^{2+}$ (Student's $t$-test, $P = 0.152$). (**b**) Current–voltage ($I$–$V$) curve of ATP-induced inward current determined by subtracting the voltage ramps (from $-130$ to $-60$ mV, 10 mV s$^{-1}$) recorded during ATP puffing from the same ramps recorded at rest. (**c**) Sample trace showing changes in the interneuron input resistance (Rin) monitored by $-50$ pA current injections every 5 s before, during and after ATP puffing. (**d**) Time course of the Rin changes in interneurons during ATP or adenosine application (two-way analysis of variance (ANOVA), $F_{(1,240)} = 68.78$, $P < 0.0001$). (**e**) Summary data showing the percentage inhibition of ATP-induced depolarization by various K$^+$ channel blockers (Student's $t$-test, XE991: $P = 0.32$, Ach: ***$P < 0.0001$, ***$P = 0.0008$, muscarine: ***$P < 0.0001$, ***$P < 0.0001$, quinidine: ***$P < 0.0001$, **$P = 0.0063$, lidocaine: ***$P < 0.0001$, **$P = 0.0069$, bupivacaine: **$P = 0.0015$, **$P = 0.0087$, pH6.4: ***$P < 0.0001$, **$P = 0.0018$). (**f**) Summary data showing the inhibition of ATP-induced depolarization by intracellular application of arachidonic acid (Student's $t$-test, ***$P < 0.0001$,). (**g,h**) Summarized data showing aCSF alkalization (pH 8.5, paired Student's $t$-test, **$P = 0.0025$, **$P = 0.0036$) and isoflurane (100 μM, paired Student's $t$-test, *$P = 0.025$, **$P = 0.0022$) enhance ATP-induced depolarization. Open squares and filled circles represent individual and averaged depolarization amplitudes, respectively. (**i,j**) Summary of ATP-induced depolarization amplitude with intracellular application of GDPβs (2 mM, Student's $t$-test, ***$P < 0.0001$), bath application of suramin (100 μM, Student's $t$-test, ***$P < 0.0001$), long-term bath application of U73122 (20 μM, Student's $t$-test, $P = 0.26$) or neomycin (100 μM, Student's $t$-test, ***$P = 0.00088$), intracellular application of heparin (10 U ml$^{-1}$, Student's $t$-test, *$P = 0.023$). (**k,l**) Sample traces and summary data showing the time course of AP firing frequency before, during and after light stimulation (bar) in the presence of quinidine (open square, two-way ANOVA, $F_{(9,167)} = 2.78$, $P = 0.0046$) or lidocaine (filled circle, two-way ANOVA, $F_{(9,177)} = 2.78$, $P = 0.0046$) in interneurons (IN). NS, not significant.

interneurons was much higher than in pyramidal neurons as reflected by fluorescence intensity (Supplementary Fig. 9). Thus, ATP may excite interneurons through inhibition of K2P channels, possibly TASK-3.

Activation of P2Y1 receptors is coupled to G$_{q/11}$, resulting in activation of phospholipase C (PLC) and subsequent hydrolysis of phosphatidylinositol 4,5-bisphosphate (PIP2; ref. 19). First, we tested whether G-proteins were involved in the ATP-induced interneuron depolarization, and found that it was robustly inhibited by intracellular application of GDP-β-S (2 mM; Fig. 5i).

Bath application of suramin (100 μM), a G-protein inhibitor[37] for >2 h also strongly inhibited the ATP effect (Fig. 5i), demonstrating that G-proteins are required for the ATP inhibition of K2P. We next tested the involvement of the subsequent signals, PLC and PIP2, and found that pre-treatment plus bath application of a PLC inhibitor, U73122 (20 μM), failed to block the ATP-induced depolarization (Fig. 5j). Similar treatment with the PIP2 scavenger neomycin (100 μM) or intracellular application of the IP3 receptor antagonist heparin (10 U ml$^{-1}$; Fig. 5j) potentiated rather than inhibited the

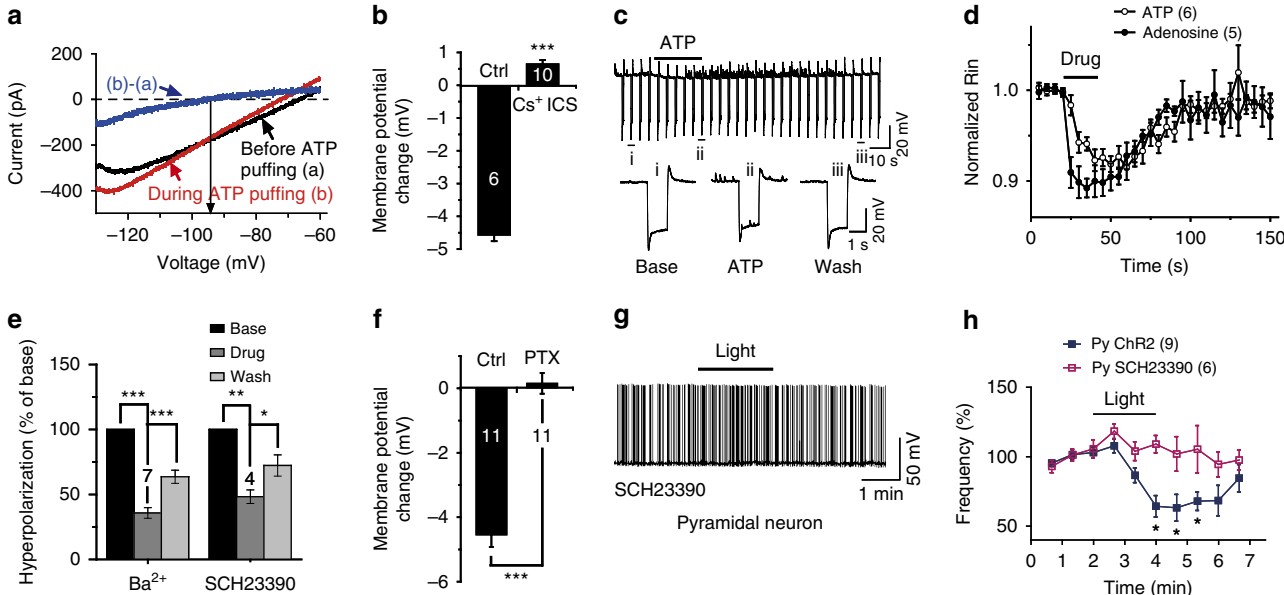

**Figure 6 | ATP inhibits pyramidal neurons by opening GIRK channels.** (**a**) Current–voltage (*I–V*) curve of ATP-induced inward current determined by subtracting the voltage ramps (from $-130$ to $-60$ mV, $10$ mV s$^{-1}$) recorded during ATP puffing from the same ramps recorded at rest. (**b**) Summary of ATP-induced membrane potential changes recorded from pyramidal neurons using K$^+$- or Cs$^+$-based ICS (Student's *t*-test, ***$P < 0.0001$). (**c**) Sample trace showing pyramidal neuron Rin recorded before, during and after ATP ($100\,\mu$M) puffing monitored by $-50$ pA current injection every 5 s. (i), (ii) and (iii) indicate representative Rin traces in each period as indicated. (**d**) Time course of interneuron Rin changes during ATP or adenosine application (two-way analysis of variance (ANOVA), $F_{(1,300)} = 5.19$, $P = 0.0235$). Data are normalized to the Rin averaged from 20 s of recording before drug application in each case. (**e**) Summary data showing the percentage inhibition of ATP-induced hyperpolarization by the GIRK channel blocker Ba$^{2+}$ ($10\,\mu$M, Student's *t*-test, ***$P < 0.0001$, ***$P = 0.00036$) or SCH23390 ($10\,\mu$M, Student's *t*-test, **$P = 0.0013$, *$P = 0.027$). (**f**) Summary of ATP-induced pyramidal neuron membrane potential change in cultured slices with or without PTX (Student's *t*-test, ***$P < 0.0001$). (**g,h**) Sample trace and summary data showing the time course of AP firing frequency before, during and after light stimulation (bar) in the presence of SCH23390 in pyramidal neurons (Py) (two-way ANOVA, $F_{(9,120)} = 2.21$, $P = 0.026$). Data are normalized to the AP frequency averaged from 2 min of recording before light stimulation in each case.

ATP-induced depolarization, suggesting that the activity of PLC to hydrolyse PIP2 is not required for the ATP-evoked inhibition of K2P. Similar results have been obtained regarding the serotonin inhibition of the TASK-3 channel[37].

**ATP inhibits pyramidal neurons by opening GIRK channels.** The reversal potential of the ATP-induced response in pyramidal neurons, determined by the same protocol as in interneurons, was $-92.3 \pm 2.58$ mV, comparable to the calculated K$^+$ equilibrium potential ($-89.4$ mV) when the extracellular K$^+$ concentration was $3.5$ mM (Fig. 6a). In addition, Cs$^+$-based intracellular solution (ICS) almost completely blocked the ATP-induced hyperpolarization (Fig. 6b), further supporting the notion that ATP inhibits pyramidal neurons by opening K$^+$ channels. In accordance with this, the input resistance of pyramidal neurons decreased during light stimuli and ATP application (Fig. 6c,d, Supplementary Fig. 7c,d).

The activation of A1 receptors is coupled to G$_{i/o}$ which results in a decrease of cAMP by the G-protein $\alpha$-subunit or opening of GIRK by the $\beta\gamma$-subunit[19]. We found that treatment of apamin, tolbutamide, bumetanide, 4-AP or ACh had no effect on ATP-induced pyramidal neuron hyperpolarization, excluding the involvement of other K$^+$ channels (Supplementary Fig. 8e). However, the GIRK channel blockers SCH23390 ($50\,\mu$M) and Ba$^{2+}$ reversibly blocked the ATP-induced hyperpolarization (Fig. 6e). It is well known that the opening of GIRK channel is mediated by the G$_{i/o}$ $\beta\gamma$-subunit. Interestingly, the ATP-induced hyperpolarization was completely blocked in hippocampal slices co-cultured with the G$_{i/o}$ inhibitor pertussis toxin for $>24$ h (Fig. 6f). Altogether, these results suggest that ATP hyperpolarizes pyramidal neurons by opening GIRK channels. Consistently, the

light-induced inhibition of pyramidal neuron firing was also abolished by SCH23390 (Fig. 6g,h). Thus, ATP differentially modulates the excitability of interneurons and pyramidal neurons through the closing and opening of two distinct types of K$^+$ channel.

**P2Y1 and A1 receptors are differentially expressed.** Because ATP has opposite effects on CCK-positive interneurons and pyramidal neurons, we used single-cell RT–PCR to investigate whether these two types of neurons have differential expression patterns of P2Y1 and A1 receptors in an independent experiment. GFP-expressing interneurons located in the CA1 SR and SLM areas of GAD-GFP mice were patched and categorized into two groups based on their responses to exogenous ATP ($100\,\mu$M). Neurons with depolarization $>2$ mV were referred to as positive, while others were regarded as negative. After electrophysiological recording, the cytoplasm was harvested and the *P2Y1*, *A1* and *GAD65* genes were amplified. We found that in the positive group, 92.9% (39 of 42) expressed the P2Y1 receptor, whereas only 35.7% (15 of 42) expressed the A1 receptor. In the negative group, only 35.9% (15 of 39) expressed the P2Y1 receptor, while 66.7% (26 of 39) expressed the A1 receptor. All the neurons examined expressed the interneuron marker GAD65 (Fig. 7a,b).

In another independent experiment, CA1 pyramidal neurons were recorded and harvested. Single-cell RT–PCR results showed that among all the pyramidal neurons tested only 3.3% (1 of 30) expressed the P2Y1 receptor, 76.7% (23 of 30) expressed the A1 receptor and all expressed pyramidal neuron marker CAMKII (Fig. 7c,d). These results explain our electrophysiological findings. Thus, interneurons, which mainly express P2Y1 receptors, are more prone to be excited by ATP, whereas pyramidal neurons,

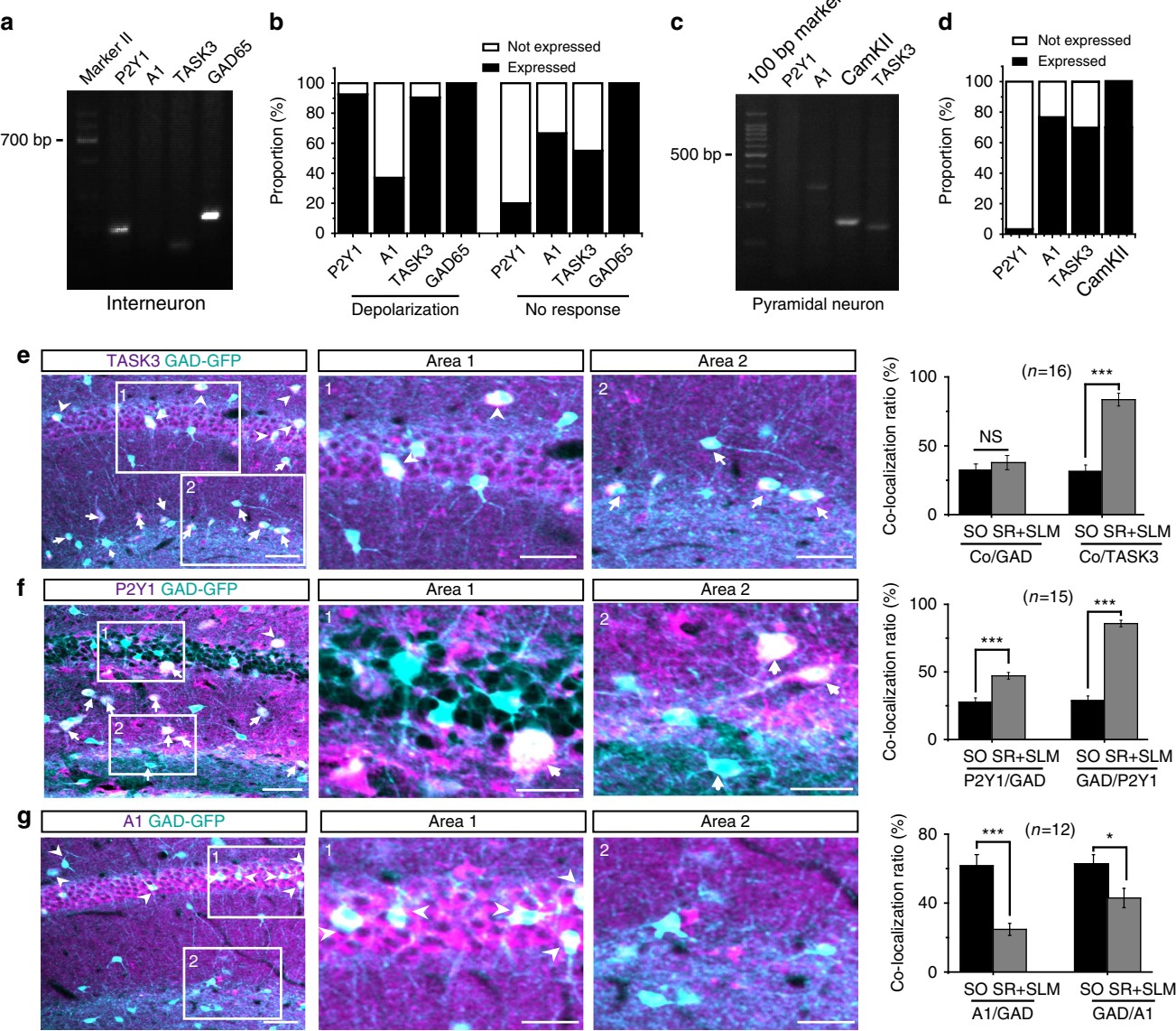

**Figure 7 | TASK3 channel and P2Y1/A1 receptors expression patterns differ between interneurons and pyramidal neurons.** (**a**) Representative agarose gel electrophoresis image of single-cell RT–PCR showing the expression of P2Y1, A1, TASK3 and GAD65 in an ATP-depolarized interneuron. (**b**) Summary data showing the percentage of P2Y1, A1, TASK3 and GAD65 expression in interneurons. (**c**) Representative agarose gel electrophoresis image of single-cell RT–PCR showing the expression of P2Y1, A1, TASK3 and CAMK II in a pyramidal neuron. (**d**) Summary data showing the percentage of P2Y1, A1, TASK3 and CAMK II expression in pyramidal neurons. (**e**) Left: immunostaining for the expression of TASK3 in GAD-GFP mouse hippocampus. Scale bars, 60 μm. Insets: enlargement of areas 1 and 2 at left. Scale bars, 30 μm. Right: summary data showing the percentage of co-localization between TASK3 and GAD-GFP in SR/SLM and stratum oriens (SO) (Student's $t$-test, $P = 0.446$, ***$P < 0.0001$). (**f**) Left: immunostaining for the expression of P2Y1 receptors in GAD-GFP mouse hippocampus. Scale bars, 60 μm. Insets: enlargement of areas 1 and 2 at left. Scale bars, 30 μm. Right: summary data showing the percentage of co-localization between P2Y1 and GAD-GFP in SR/SLM and SO (Student's $t$-test, ***$P < 0.0001$, ***$P < 0.0001$). (**g**) Left: immunostaining for the expression of A1 receptors in GAD-GFP mouse hippocampus. Scale bars, 80 μm. Insets: enlargement of areas 1 and 2 at left. Scale bars, 40 μm. Right: summary data showing the percentage co-localization of A1 and GAD-GFP in SR/SLM and SO (Student's $t$-test, ***$P < 0.0001$, *$P = 0.0144$,). Note that numbers indicate slices being analysed not cell numbers. Co-localization is indicated by arrows in the SR/SLM and by arrowheads in the SO/stratum pyramidale (SP). NS, not significant.

which express mainly A1 receptors, tend to be inhibited by ATP. This conclusion was further supported by the immunostaining results. When slices from GAD-GFP mice were stained for TASK3, P2Y1 and A1 antibodies, we found that TASK 3 highly expressed in interneurons (Fig. 7e) and P2Y1 co-localized with GAD-GFP much more frequently in the SR/SLM than in the stratum oriens; this was consistent with the results obtained from electrophysiological recordings that most ATP-excited interneurons were located in the SR/SLM (Fig. 7f). In contrast,

A1 only co-localized with GAD-GFP occasionally, whereas robust A1 expression was found in the stratum pyramidale (Fig. 7g).

**Light activation of astrocyte modulates network activity.** To further explore the physiological relevance of astrocyte-derived ATP, we examined the effects of light activation of astrocytes on kainate-induced *ex vivo* gamma oscillations in the hippocampus slice, an *in vitro* model to study rhythmic brain activity which

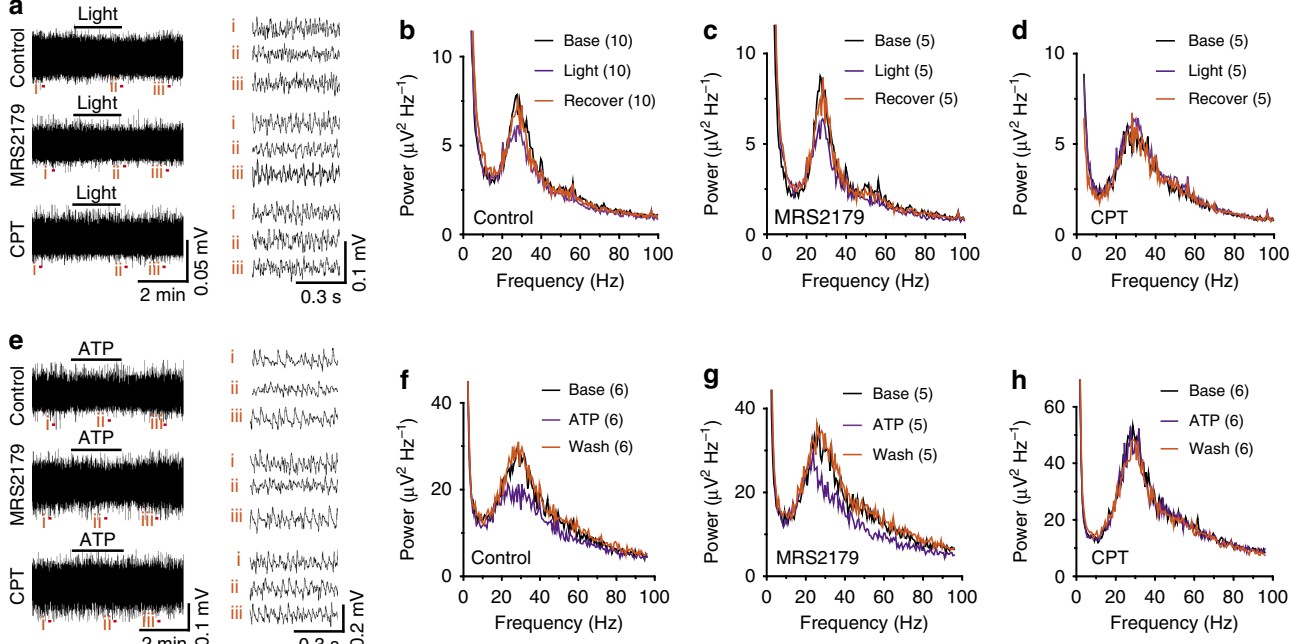

**Figure 8 | Astrocyte-derived ATP downregulates the power of kainic acid-induced *ex vivo* gamma oscillation *in vitro*.** (**a**) Sample traces of kainic acid-induced *ex vivo* gamma oscillation before, during and after 2 min light stimulation (1 Hz, 0.5 s on and 0.5 s off) of astrocytes expressing ChR2 in control conditions (top), in the presence of 20 μM MRS2179 (middle), and in 1 μM CPT (bottom). (i), (ii) and (iii) indicate representative traces in each period. (**b–d**) Summary power spectra of kainate-induced gamma oscillation before (black), during (violet) and after (orange) light stimulation in control conditions, in the presence of 20 μM MRS2179, and in 1 μM CPT. (**e**) Sample traces of kainic acid-induced *ex vivo* gamma oscillation before, during and after application of 1 μM ATP in control conditions (top), in the presence of 20 μM MRS2179 (middle), and in 1 μM CPT (bottom). (i), (ii) and (iii) indicate representative traces in each period. (**f–h**) Summary power spectra of kainate-induced *ex vivo* gamma oscillation before (black), during (violet) and after (orange) application of 1 μM ATP in control conditions, in the presence of 20 μM MRS2179, and in 1 μM CPT.

may have an important role in learning, memory and cognition[45]. We found that light activation of ChR2-expressing astrocytes decreased the power of *ex vivo* gamma oscillation induced by bath application of 100 nM kainic acid (Fig. 8a,b), an effect that was mimicked by bath application of 1 μM ATP (Fig. 8e,f). Furthermore, the decreased *ex vivo* gamma oscillation power induced by either light stimulation or bath application of ATP was blocked by the A1 receptor antagonist CPT, but not by the P2Y1 receptor antagonist MRS2179 (Fig. 8). These results indicated that the ATP-induced decrease in the power of gamma oscillation was mainly mediated by direct inhibition of pyramidal neuronal activity, consistent with previous reports that CCK-positive interneurons are not involved in the modulation of gamma oscillation[46–48].

## Discussion

Accumulating evidence suggests that astrocytes modulate neuronal excitability through the release of gliotransmitters. It has been reported that Müller cell-derived ATP decreases the excitability of retinal neurons[17]. Astrocytes in the brainstem chemoreceptor area release ATP to excite chemoreceptor neurons and trigger respiratory responses *in vivo*[15]. We discovered that astrocyte-derived ATP increased interneuron activity by closing K2P channels through the activation of P2Y1 receptors and decreased pyramidal neuron activity by opening GIRK channels through the activation of A1 receptors in the hippocampal CA1 area. By simultaneously increasing the excitability of inhibitory neurons and decreasing the excitability of excitatory neurons, the whole network activity would be downregulated after elevated astrocyte activity. Our results indicate that once the gliotransmitters are released they may not be restricted to the peri-synaptic region, but diffuse into extracellular space and activate receptors on

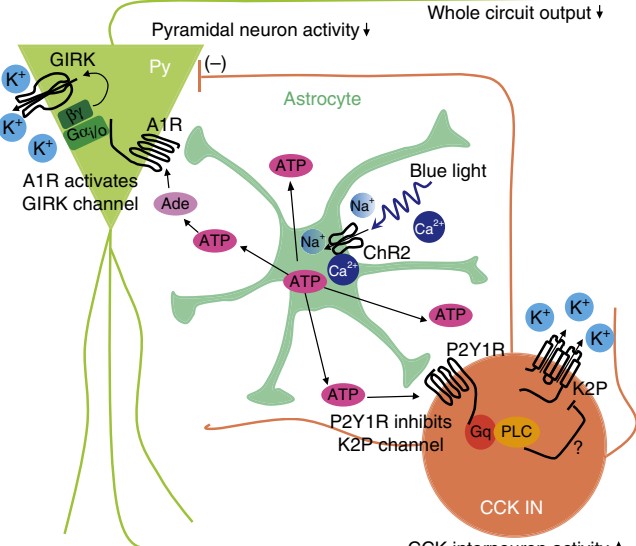

**Figure 9 | Working model of astrocytes inversely modulates the excitability of pyramidal (Py) neurons and interneurons.** Light activation of ChR2-expressing astrocyte (centre) induces ATP release, which directly increases the excitability of CCK[+] interneurons (right) through P2Y1 receptor-mediated inhibition of K2P potassium channels. The astrocyte-released ATP can be degraded into adenosine (Ade), which in turn decreases the excitability of pyramidal neurons (right) through A1 receptor-mediated activation of GIRK channels. The ATP-induced excitation of CCK[+] interneurons will further inhibits pyramidal neuron activity through GABAergic synapses. In this way, the astrocyte-derived ATP efficiently regulates the output of whole neuronal circuit.

nearby neurons[49]. Moreover, gliotransmitters may also be released near neuronal dendrites and somata where astrocytes also make extensive close contacts[49,50].

Previous studies assumed that, under physiological conditions, extracellular ATP is mainly degraded into adenosine to perform its actions, due to the high expression levels of ectonucleotidases[11,13]. However, recent studies have shown that the concentration of extracellular ATP is high enough to activate P2 receptors after relatively strong stimulation[10,15,17]. In this study, we also found that the ATP itself directly activates CCK-positive interneurons. Homeostasis may exist between the release and hydrolysis of ATP to maintain the extracellular ATP at a specific level and tonically increase interneuron excitability.

We demonstrated that the reason why astrocyte-derived ATP has opposite effects on the excitability of interneurons and pyramidal neurons is the differential expression of purine receptor subtypes. Our results showed that CCK-positive interneurons mainly express P2Y1 receptors directly coupled to K2P channels and mediate ATP-induced depolarization. In contrast, pyramidal neurons mainly express A1 receptors that gate GIRK channels through $G_{i/o}$ $\beta\gamma$-subunits and mediate ATP/adenosine-induced hyperpolarization. P2Y1 receptors mediate excitatory signals, while A1 receptors process inhibitory signals[19,51]. By acting on excitatory P2Y1 receptors in inhibitory neurons and inhibitory A1 receptors in excitatory neurons, ATP inhibits the activity of the whole hippocampal CA1 network (Fig. 9). Furthermore, both P2Y1 and A1 receptors have more than one downstream effector, so it is plausible that integration of these effectors determines the final effect of ATP on neuronal excitability[8,9,11,19].

Interneuron activity is critical for neuronal network functions. The connectivity, transmitter release and intrinsic membrane properties of interneurons are finely tuned to control the activity patterns of the circuit. Hippocampal interneurons are diverse and can be subdivided into several types based on their physiological properties, neurochemical markers and the anatomical distribution in the circuit[34,35]. CCK-positive and PV-positive basket cells are major perisomatic inhibitory interneurons that control population discharge patterns, and therefore all the higher brain functions executed by the hippocampus. Electrophysiological and electron-microscopic studies show that PV cells receive several times more excitatory glutamatergic inputs than CCK cells[52,53]. Our findings that astrocyte-derived ATP specifically increases the excitability of CCK cells, but not that of PV cells, through inhibiting K2P channels, indicate that the excitability of the two types of perisomatic inhibitory interneurons are differentially modulated through distinct mechanisms. Thus, although CCK cells receive less glutamatergic excitation, their excitability could be maintained by astrocyte-derived ATP[47,48]. Interestingly, recent studies reported that astrocyte-derived ATP[54] and K2P channel blockers[55] have antidepressant effects.

The light-induced $Ca^{2+}$ elevation in ChR2-expressing astrocytes is mediated by $Ca^{2+}$ influx from extracellular space[10,26,56] and may propagate to the surrounding astrocytes[10], probably through intercellular gap junctions, which allow astrocytes to function as a network[2,4]. ChR2 has a high permeability to $Na^+$ (ref. 26) and the $Na^+/Ca^{2+}$ exchanger has been found to mediate ChR2-induced $Ca^{2+}$ elevation in astrocytes[56]. Interestingly, $Ca^{2+}$ influx through plasma membrane mediated by TRPC and TRPA1 channels has been reported recently essential for gliotransmitter release and synaptic modulation[57–59]. Thus, ChR2-mediated $Ca^{2+}$ influx through plasma membrane may nicely mimic astrocyte $Ca^{2+}$ signalling in various physiological and pathological conditions[56,60].

Prolonged astrocyte activity may be induced by persistent neural activity that occurs in both physiological (such as synaptic plasticity associated with learning and memory) and pathological conditions (such as fear, stress, seizure and stroke). By inhibiting pyramidal neurons and exciting interneurons, astrocytes may have a critical role in stabilizing neuronal network activity under physiological conditions and protect the hippocampus from overexcitation under pathological conditions such as epilepsy and stroke.

## Methods

**Slice preparation.** All animal procedures were approved by Shanghai Institutes for Biological Sciences Animal Research Advisory Committee, Chinese Academy of Science. Mice (3–4 weeks postnatal) were anaesthetized with 1% sodium pentobarbital, then the blood was replaced by cardiac perfusion with ice-cold oxygenated (95% $O_2$ and 5% $CO_2$) artificial cerebrospinal fluid (aCSF) before decapitation, after which the whole brain was removed rapidly into ice-cold oxygenated aCSF containing (in mM): 110 choline Cl, 3.5 KCl, 0.5 $CaCl_2$, 7 $MgCl_2$, 1.3 $NaH_2PO_4$, 25 $NaHCO_3$ and 20 glucose. After the hippocampal formation was dissected, transverse slices (250–300 μM) were cut on a vibratome (HM 650 V, Microm) and allowed to recover at 32 °C for 1 h before recording. The aCSF used for recovery and recording contained (in mM): 125 NaCl, 3.5 KCl, 2 $CaCl_2$, 1.3 $MgCl_2$, 1.3 $NaH_2PO_4$, 25 $NaHCO_3$ and 10 glucose. For experiments, an individual slice was transferred to a submerged recording chamber and continuously perfused with the above aCSF (3.0 ml min$^{-1}$) at 26 °C or 32 °C as indicated. The slices were visualized under a microscope (DMLFS, Leica) using infrared video microscopy and differential interference contrast optics.

**Electrophysiology.** Interneurons and pyramidal neurons in the hippocampal CA1 area were identified based on their location, shape and firing properties. The patch electrodes were made from borosilicate glass capillaries (B-120-69-15, Sutter Instruments) with a resistance in the range of 3–5 MΩ. The internal solution contained (in mM): 125 K-gluconate, 15 KCl, 10 HEPES, 4 $MgCl_2$, 4 $Na_2ATP$, 0.4 $Na_3GTP$, 10 Tris-phosphocreatine and 0.2 EGTA. In some cases, $K^+$ was replaced by the same concentration of $Cs^+$ to block $K^+$ channels intracellularly. Recordings were made with an Axon 700A patch-clamp amplifier and 1320A interface (Axon Instruments). The signals were filtered at 2 kHz using amplifier circuitry, sampled at 10 kHz and analysed using Clampex 9.0 (Axon Instruments).

**Viral injection and optical methods.** The hGFAP-Cre mouse with Cre recombinase driven by human GFAP promoter was a gift from K.D. McCarthy's laboratory[61]. The genetic background of the mice was C57BL/6. ChR2 virus was constructed as described previously[10]. Vector bearing EGFP was used as negative control. Mice (18 days postnatal) were anaesthetized with 1% sodium pentobarbital and fixed in a stereotaxic apparatus. A small hole in the skull was made using a dental drill, 1.8 mm from the midline and 1.3 mm anterior to the posterior fontanelle. The needle (Hamilton Instruments) was lowered to 2.3 mm below the dura, left there for 5 min before retraction 0.5 mm toward the surface, and then 2 μl virus was injected at 0.2 μl min$^{-1}$ using a microsyringe pump (Stoelting Instruments). After injection, the needle was held in place for another 5 min before complete removal from the brain. The experiments were carried out at least 7 days after injection. Photostimulation was delivered by 473-nm solid-state laser diodes, and light pulses were generated with a custom-built high-speed shutter; the power density of the blue light was 8–12 mW mm$^{-2}$, measured with a power meter (Coherent Instruments). Blue light pulses (500 ms) at 1 Hz were delivered to the slices through a quartz fibre (200 μm diameter, custom-made) for 2 min; the estimated size of the projection area of the photostimulation onto the slice was 2–3 mm$^2$. After recording, the slices were fixed in 4% paraformaldehyde (PFA) and immunostained with anti-RFP antibody (1:100, rat monoclonal, ChromoTek) to highlight the expression of ChR2.

**Immunohistochemistry and confocal imaging.** P21-28 *GAD-GFP* or *C57BL/6* mice were anaesthetized with 1% sodium pentobarbital and perfused transcardially with normal saline (0.9% NaCl) followed by PFA (4%) dissolved in phosphate-buffered saline (PBS, pH 7.4). The brain was removed and post-fixed in 4% PFA for 4–6 h and then immersed in 30% sucrose containing PBS until the brain settled (overnight at 4 °C). Coronal sections (30 μm) were cut on a cryostat microtome (Leica CM1900). The sections were treated with 0.2% Triton X-100 for 30 min and blocked in 10% bovine serum albumin (BSA) for 1 h. Slices from *GAD-GFP* mice were stained with anti-P2Y1 antibody (1:200, polyclonal, Abcam) or anti-A1 antibody (1:50, polyclonal, Santa Cruz) for 24 h at 4 °C. In some electrophysiological experiments, recorded interneurons were actively filled with biocytin by current injection for up to 15 min (+500 pA for 1 s at 0.2 Hz). After recording, the slices were fixed and immunostained with anti-CCK antibody (1:100, monoclonal, CURE). Biocytin-labelled interneurons were visualized with streptavidin-conjugated Alexa-488 (1:1,000, Molecular Probes). The sections were imaged under a confocal microscope (IX71, Olympus) with Fluoview 500 using a × 20/0.7

objective lens (Olympus). The scans from each channel were collected in multitrack mode to avoid crosstalk between channels.

**Calcium fluorescence imaging.** Slice imaging was performed using a confocal laser scanning microscope (Nikon A1R) with a Nikon $\times 20/0.8$ NA water immersion objective. The astrocytes were bulk loaded with Fluo-4 (1 mM) or Rhod-2 (1 mM). Alexa Fluor-647 (30 µM) and BAPTA (15 mM) were loaded through the whole-cell patch pipette, and small depolarizing current pulses (5 mV depolarization, 50 ms duration, 10 Hz) were injected into the cell to facilitate BAPTA diffusion. The sampling rate was 0.9 Hz. Relative changes in fluorescence were calculated and normalized ($\Delta F/F$).

**Measurement of extracellular ATP concentration.** An enzyme-based ATP biosensor was used to measure the concentration of extracellular ATP (Sarissa Biomedical, UK). The sensitive part of the sensor was $\sim 2$ mm in length. Before measuring, a standard curve was generated using standard ATP samples. Then the same ATP sensor and a null sensor were placed closely on the surface of the slice (within 100 µM). The ATP sensor generated current through hydrolysing ATP, and the signal was detected by a potentiostat (micro C, WPI), then the signal was digitized by LabChart (ADinstrument, USA) for further analysis.

***In vitro* local field potential recording.** The slices used for *in vitro* extracellular field potential recording were prepared as usual. Recordings were made in the stratum pyramidale of CA1 using glass microelectrodes containing aCSF (resistance 3–5 MΩ) in submerged-type chambers with a self-made insert to ensure both sides of the slice were well supplied with oxygen[62]. Kainic acid (100 nM) was added to the perfusate to induce gamma oscillation. The data were recorded with a MultiClamp 700A amplifier 1320A interface (Axon Instruments). Fast Fourier transformations for power spectra were computed from 60-s traces using Axograph software. Power values were derived from integrating the power spectra between 5 and 100 Hz.

**Single-cell RT-PCR.** The neuronal expression patterns of TASK3, P2Y1 and the A1 receptor, as well as the identity of interneurons, were examined using the single-cell RT-PCR. The SuperScript III CellsDirect cDNA Synthesis System kit (Invitrogen, Carlsbad, CA, USA) was used to synthesize first-strand cDNA. Pipettes with large tips (resistance 1–2 MΩ) were filled with RNase-free internal solution. After neuronal firing patterns and their responses to exogenous ATP had been recorded, the cytoplasm was harvested into the patch pipette and expelled into an RNase-free PCR tube containing re-suspension buffer provided with the kit. First-strand cDNA was synthesized according to the kit description using gene-specific antisense primers (Supplementary Table 1). The RNA template was removed by digestion with RNase H for 10 min at 37 °C. Subsequently, two rounds of PCR amplification reactions were performed using rTaq polymerase (Takara) with a programmable thermal cycler (Bio-Rad). In the first round, all the target genes were amplified together by pooling all the pairs of outer primers in one tube. After that, the PCR product was divided into several aliquots and each gene was amplified individually by inner primers. PCR products were separated by electrophoresis in 3% agarose gels and the images were captured on a gel imaging system (Tanon 2500). The identities of all PCR products were confirmed by sequencing.

**Statistics.** Data are presented as mean ± s.e.m., and statistical comparisons were assessed with the two-way analysis of variance and paired or unpaired Student's *t*-test as appropriate. For grouped time point data, first two-way analysis of variance was used to compare difference between groups, then Student's *t*-test was used for individual time point comparison. $P < 0.05$ was taken as significant.

**Data availability.** All the relevant data are available from the authors.

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

## Acknowledgements

We thank Professor I.C. Bruce for discussions and reading the manuscript, G. Ohning for providing the CCK monoclonal antibody, K.D. McCarthy for providing the hGFAP-Cre mice, K. Deisseroth for the gift of humanized ChR2 plasmid and Z.J. Huang for the gift of ChR2-mcherry floxed with a STOP cassette. This work was supported by grants from the Major State Basic Research Program of China (2016YFA0501000), the National Natural Science Foundation of China (31190060, 31490590, 91132000, 81221003, 91232000), the National Key Technology R&D Program of the Ministry of Science and Technology of China (2012BAI01B08), 111 Project (B13026) and the Fundamental Research Funds for the Central Universities (2014FZA7007).

## Author contributions

S.D. and Z.T. conceived the idea, analysed the data and wrote the manuscript; Z.T. and Y.L. carried out most of the experiments; L.M. gave suggestions and provided reagents; W.X. helped with electrophysiological recordings; H.L., L.Z. and Z.G. helped with mouse breeding.
