## [Peer Review File · Nature Communications]

Reviewers' comments:

Reviewer #1 (Remarks to the Author):

In the present manuscript authors have investigated the effects of astrocytes activation on the excitability of hippocampal pyramidal neurons and interneurons. They have combined electrophysiological recordings in hippocampal slices, optogenetic stimulation of astrocytes, pharmacological analysis, and immunocytochemistry. They claim that channelrhodopsin activation of astrocytes induces the release of ATP that activate different purinergic ATP receptors in pyramidal cells and CCK+ interneurons, which lead to the decrease and increase their excitability, respectively. This is an interesting and study that may add valuable information regarding a relevant current topic in neuroscience, i.e., the regulatory role of astrocytes in neuronal activity.

The authors have investigated an interesting idea, and provide novel and potentially interesting results. Some of the conclusions reached are sound, but others, and probably the most important ones, are insufficiently supported by present results. Indeed, as detailed below, there are several important concerns, especially related to the methodology used and the experimental design that need to be addressed to make the conclusions convincing.

Moreover, while the pharmacological analysis of purinergic signaling is adequate and exhaustive, the major claim stated in the title of the manuscript, that is, that this purinergic signaling derives from astrocytic activity is weakly supported by experimental results.

Specific comments:

1. Based on a previous report of the group (Chen et al. *Glia* 2013), authors assume that channelrhodopsin (ChR2) effects are mediated by calcium elevations in astrocytes (Pg 5). However, this is an important piece of evidence that should be shown in the present manuscript.
2. Related to the previous point, temporal correlations between astrocyte calcium elevations and neuronal excitability changes should be performed. In contrast to synaptic transmission studies where synaptic inputs cannot be spatially discerned, the analysis of neuronal excitability (mainly derived from somatic or proximal dendrites) makes feasible the correlation between neuronal excitability changes and calcium levels in astrocytes close to perisomatic areas.
3. The stimulus duration and the delayed and slow time course of the responses is a concern. The analysis of the calcium dynamics evoked by ChR2 may help to understand why it is necessary to stimulate for 2 min with blue light, why there is a delay of more than 2 min from the stimulation onset to observe the effects and why the responses lasted several minutes after the cessation of the stimulus.
4. One of the major concerns derived from the claimed contribution of astrocytes to the differential ATP effect in pyramidal neurons and CCK interneurons. In the Fig authors conclude that astrocytes release ATP because ChR2 effects are mimicked by direct application of ATP. This is certainly a good control, but it is not sufficient to grant the authors' experimental design of performing many important pieces of the study using direct application of ATP instead of direct astrocyte stimulation. Indeed, differential effects may simply reflect different receptor activation by directly applied ATP, but not necessarily by ATP released from astrocytes stimulated with ChR2.
5. On page 5, authors state "To determine whether these responses were Ca²⁺-dependent, we loaded BAPTA into astrocytes...". Then, they report that after loading astrocytes with BAPTA, no neuronal changes occurred. However, the effectiveness of BAPTA loading should be confirmed, by monitoring calcium levels before and after BAPTA loading. Moreover, in addition to ChR2, an additional agonist should be used to test that. This is an important control that needs also to be shown.

6. Authors assessed the effects of ChR2-stimulation of astrocytes analyzing the spontaneous firing rates of the recorded neurons. This parameter is subject to multiple variables (e.g., it is very sensitive to small changes in resting potential, internal solution, intracellular rundown mechanisms, etc). The reported experiments directly show changes in neuronal excitability, but standard procedures more accurate and well established in the literature needs to be used to support the validity of the conclusion. Specifically, recordings in voltage-clamp conditions monitoring changes in the membrane current and the membrane conductance.

Authors have adequately done that when puffing ATP (e.g., Fig. 3A-C), but this should also be done in other experimental approaches and, more importantly, in ChR2 experiments.

7. Moreover, ATP is known to elevate calcium in hippocampal astrocytes, which may lead to the release of other gliotransmitters, including ATP. This may affect the pharmacological analysis performed, because the sensitivity of the neuronal effects to purinergic antagonists may be due to indirect blockade of astrocytic receptors.

8. ATP is known to regulate hippocampal synaptic transmission. At the beginning of the manuscript (Pg 5), it is stated that experiments were performed in the presence of ionotropic glutamate and GABA receptor antagonists. Authors should clarify whether this is the case for this initial experiments or apply to the rest of the manuscript (except when monitoring synaptic currents). I assume that is the case, otherwise it would complicate the possible interpretations by adding synaptic effects. Authors should clearly state the experimental conditions that apply.

9. In Pg 6 authors state that "The onset of the responses of exogenous ATP is much faster than light stimulation, reflecting the feature of ATP exocytosis from astrocytes". As indicated above, since calcium levels were not analyzed, it remains unknown whether the different delays are due to the feature of ATP exocytosis or a delayed upstream mechanism such as delayed calcium elevations. Once again, the analysis of the astrocyte calcium responses is necessary.

10. In most of the pharmacological experiments, authors claim that the ATP-induced effect is blocked. However, the responses are significantly reduced but not completely abolished. These partial blockage suggest additional mechanisms in addition to the simple interpretation provided. How authors interpret such partial reductions?

11. In Pg 22, the discussion of the mechanisms of both ChR2 stimulation and gliotransmitter release is too speculative and greatly superficial. Authors could contribute to clarify these issues by analyzing the astrocyte calcium evoked by ChR2 and its source.

Reviewer #2 (Remarks to the Author):

This manuscript describes the differential role of astrocytic ATP in CCK positive interneurons and CA1 pyramidal neurons in mouse hippocampus. Authors claim astrocytic ATP excites CCK positive interneurons via P2Y1 and by inhibiting K2P channels while ATP (after degradation by ectonuclease to adenosine) inhibits pyramidal neurons via A1 and activating GIRK channels.

Although the study is interesting and loaded with massive amount of data, there are several important issues to be resolved and clarified.

1. The use of GFAP-cre mouse is known to be problematic due to expression of GFAP in neuronal progenitor cells.

2. In Fig. S1D, authors need to count more than 15 or 19 cells (perhaps over 100 cells) to make sure that ChR2-mCherry is expressed only in astrocytes, but not in neurons or NG2 positive cells.
3. The materials and method should contain at least minimal information regarding the virus and mouse information regarding how ChR2-mCherry virus was constructed, introduced into mouse brain, and where it was injected, instead of simply referring to Supplementary Info.
4. Use of ChR2 for Ca²⁺ entry is known to be problematic due to its low Ca²⁺ permeability and undesirable changes of intracellular pH. (See, Optogenetic counteracting of glial acidosis suppresses glial glutamate release and ischemic brain damage. Beppu K, Sasaki T, Tanaka KF, Yamanaka A, Fukazawa Y, Shigemoto R, Matsui K. Neuron. 2014 Jan 22;81(2):314-20.)
5. In page 5, the use of EGFP for control light stimulation needs to be described in the materials and method or results section.
6. Repetitive depolarization of patched astrocyte in order to spread the BAPTA quickly is nice but this method can result in undesirable changes in physiology of astrocytes.
7. In page 6, authors need to qualify the statement, "The onset of the responses of exogenous ATP is much faster than light stimulation" by measuring the rise time and comparing the values.
8. In page 7, authors need to quantify the frequency and amplitude of sPSPs.
9. In Fig. 2A, the depolarizing current-induced AP's and firing pattern and ATP response in the same cell should be displayed. Each example of regularly spiking- and fast spiking-interneuron should be displayed.
10. In page 12, lidocaine and quinidine are known to affect AP, especially lidocaine. However, Fig. 4K (Fig. 4E as well) shows no sign of inhibition of AP firing in the presence of lidocaine or quinidine. How is this possible?
11. In page 12, the evidence TASK-3 is very weak and circumstantial. Authors should either tone down the TASK3 in abstract or directly demonstrate by using shRNA or KO mouse.
12. At first, Fig. 5A appears to be an I-V relationship, but it is not. Because readers are used to looking at I-V relationship, Fig. 5A is very confusing. For example, Fig. 4B is a typical I-V relationship. It is difficult to understand why authors decided to show Fig. 5A not as an I-V relationship.
13. In page 14, bumetanide is misspelled as "bumitanide."
14. In page 16, it is interesting to see the endogenous tonic level of ATP in brain slices. However, it is difficult to conclude that it is coming from astrocytes without further evidence. Therefore, I suggest that authors should either take out this part (Fig. 7) from this study to save for future paper or perform additional experiments to demonstrate that it is coming from astrocytes.
15. In Fig. 8, it is interesting to see the possible involvement of astrocytic ATP in gamma oscillation. However, I realized that the experiment was done on acute brain slices, but not in whole intact animal. It will be difficult to say that gamma oscillation observed in brain slices would be the same gamma oscillation in intact animals. Therefore, I suggest to call this gamma oscillation with a different name, such as "gamma oscillation in brain slice" or "ex vivo gamma oscillation" to distinguish it from

real gamma oscillation that we observe in intact animals. This word needs to be replaced in entire manuscript.

16. The discussion is too long and verbose. It needs to be shortened. Especially the section on interneuron subtypes and characterization needs to be shortened.

Reviewer #3 (Remarks to the Author):

In this manuscript, the authors have investigated the regulation of K channels by ATP released from astrocytes. There have been previous reports suggesting extracellular ATP regulated K⁺ channels, but none of these studies were performed under normal receptor and channel expression levels in brain slices. The general interest in this paper would be high, given the novel discovery of another mechanism by which astrocytes regulate neuronal excitability. The basic strategy that the authors used to approach this question in vivo was to conditionally (GFAP-Cre) express a light-activated channelrhodopsin in astrocytes. They found increased excitability of CCK interneurons (via K_{2P} by P2Y₁Rs) and decreased excitability of pyramidal neurons due to opening of GIRK via activation of A₁ receptors.

The manuscript is reasonably well-written, although it could use another pass of editing to make the dense text easier to read. The data presented in the figures are impressive, convincing and the primary conclusions presented by the authors are supported. The Authors do rely heavily on pharmacology for their primary conclusions, but did utilize the P2Y₁R KO mouse to strengthen their findings. The potential involvement of the newly defined two-pore domain K channel is very interesting, specifically the TASK family in the hippocampus. Evidence for this statement, again, is primarily based on pharmacological profiles, but their data are consistent and supportive. Impressive immunocytochemistry data are also supportive of the necessary receptors and channels being in the appropriate cell.

ATP clearly induces a hyperpolarization and its K⁺ channel based on reversal potentials. GIRK channel blockers point to its regulation being directly mediated by the beta-gamma subunit (as well as PTX). This same pharmacological profile was repeated when ATP was specifically released from astrocyte activation.

As noted above, the main text is very dense reading because of all the pharmacology. A final diagram/cartoon of all the major players (cells, channels and receptors) would be very welcome and make the arguments in the manuscript much easier to follow.

A few minor issues:

Figure legend of S1, D and E are not correct, D-F in figure misidentified. Please reference the statement "passive membrane properties expected of astrocytes".

In the text, p7, line 4, the authors state "The decreased sPSPs may reflect the direct inhibitory effect of ATP on synaptic transmission through presynaptic A₁ receptors". Authors need to reference this statement.

p. 11, their statement, based on pharmacology only, "M-current inhibition is not involved in the ATP-evoked increase in interneuron excitability" is too strong, please soften statement.

Recommend Accept with minor revision

Reviewers' comments:

Reviewer #1 (Remarks to the Author):

In the present manuscript authors have investigated the effects of astrocytes activation on the excitability of hippocampal pyramidal neurons and interneurons. They have combined electrophysiological recordings in hippocampal slices, optogenetic stimulation of astrocytes, pharmacological analysis, and immunocytochemistry. They claim that channelrhodopsin activation of astrocytes induces the release of ATP that activate different purinergic ATP receptors in pyramidal cells and CCK+ interneurons, which lead to the decrease and increase their excitability, respectively.

This is an interesting and study that may add valuable information regarding a relevant current topic in neuroscience, i.e., the regulatory role of astrocytes in neuronal activity.

The authors have investigated an interesting idea, and provide novel and potentially interesting results. Some of the conclusions reached are sound, but others, and probably the most important ones, are insufficiently supported by present results. Indeed, as detailed below, there are several important concerns, especially related to the methodology used and the experimental design that need to be addressed to make the conclusions convincing.

Moreover, while the pharmacological analysis of purinergic signaling is adequate and exhaustive, the major claim stated in the title of the manuscript, that is, that this purinergic signaling derives from astrocytic activity is weakly supported by experimental results.

Specific comments:

1. *Based on a previous report of the group (Chen et al. Glia 2013), authors assume that channelrhodopsin (ChR2) effects are mediated by calcium elevations in astrocytes (Pg 5). However, this is an important piece of evidence that should be shown in the present manuscript.*

Following the reviewer's suggestion, we performed calcium imaging experiments and confirmed that light stimulation induced calcium elevation in ChR2-expressing astrocytes, which was blocked by intracellular loading-BAPTA in astrocytes, a result consistent with light-induced excitability changes in neurons. These data have been summarized into two new figures (Fig. 2 and Fig. S5) and 3 supplemental movies in the revised manuscript.

2. *Related to the previous point, temporal correlations between astrocyte calcium elevations and neuronal excitability changes should be performed. In contrast to synaptic transmission studies where synaptic inputs cannot be spatially discerned, the analysis of neuronal excitability (mainly derived from somatic or proximal dendrites) makes feasible the correlation between neuronal excitability changes and calcium levels in astrocytes close to perisomatic areas.*

As we described above, we performed calcium imaging experiments and found that the response latency and time course of the increased calcium signal in astrocytes (Fig. 2) correlated well with that of increased extracellular ATP (Fig. 1 F) and neuronal excitability changes (Fig 1). Thus, we concluded that astrocyte calcium elevations and neuronal excitability are temporally correlated.

On the other hand, we feel it is unpractical to directly examine the (spatial) correlation between neuronal excitability changes and astrocytic calcium levels through simultaneous electrophysiological recording in neurons and calcium imaging in astrocytes, for the following reasons: (a). The excitability changes recorded in cell body may also reflect dendritic activity which may be affected by

surrounded astrocytes. (b). ATP may be released from discrete fine processes of astrocytes, then diffused to recorded neuronal cell body. The fine processes are difficult to be identified by calcium imaging. (c). Calcium dye will buffer free calcium in astrocyte which would reduce light-induced ATP release. We thus did not address the spatial correlation by simultaneously perform calcium imaging and electrophysiological recording at the same time.

3. *The stimulus duration and the delayed and slow time course of the responses is a concern. The analysis of the calcium dynamics evoked by ChR2 may help to understand why it is necessary to stimulate for 2 min with blue light, why there is a delay of more than 2 min from the stimulation onset to observe the effects and why the responses lasted several minutes after the cessation of the stimulus.*

As described above, we performed additional calcium imaging experiments and indeed found that the latency and time course of the light-induced calcium elevation in astrocytes correlated well with that of the light-induced excitability changes in neurons (Fig. 2). To more directly address this question we also monitored the concentration change of extracellular ATP during the light stimuli. This was achieved by using an ATP specific biosensor. We found that the extracellular ATP level increased $0.93 \pm 0.13 \mu\text{M}$ after light stimuli, and reached to its peak level in approximate 2 minutes after the cessation of light stimuli (Fig. 1 F, Fig. S3). Thus, the light-induced ATP release is also temporally correlated well with neuronal excitability changes. These data have been incorporated into Figure 1 and 2 in the revised manuscript.

4. *One of the major concerns derived from the claimed contribution of astrocytes to the differential ATP effect in pyramidal neurons and CCK interneurons. In the Fig authors conclude that astrocytes release ATP because ChR2 effects are mimicked by direct application of ATP. This is certainly a good*

control, but it is not sufficient to grant the authors' experimental design of performing many important pieces of the study using direct application of ATP instead of direct astrocyte stimulation. Indeed, differential effects may simply reflect different receptor activation by directly applied ATP, but not necessarily by ATP released from astrocytes stimulated with Chr2.

We agree that effects of directly applied ATP may not totally mimic that of ATP released from astrocytes induced by light-stimulation. However, our major conclusion that contribution of astrocytes to the differential ATP effects on pyramidal neurons and CCK interneurons is not only supported by the evidence that Chr2 effects are mimicked by direct application of ATP, but also supported by more direct evidence including: (a) differential effects of light stimulation on excitabilities of pyramidal neurons and CCK interneurons can be blocked by antagonists or knock out of A1 receptors and P2Y receptors, respectively (Fig. 4); (b) P2Y1 and A1 receptors and their downstream signal molecules were differentially expressed on interneurons and pyramidal neurons (Fig. 7). Thus, all the critical evidence obtained by Chr2 stimulation is supported by multiple results using different approaches. Furthermore, the result that Chr2 effects can be mimicked by direct application of ATP excludes the possible involvement of other type of glia transmitters in the light-induced differential effects on the excitabilities of pyramidal neurons and CCK interneurons.

5. *On page 5, authors state "To determine whether these responses were Ca²⁺-dependent, we loaded BAPTA into astrocytes...". Then, they report that after loading astrocytes with BAPTA, no neuronal changes occurred. However, the effectiveness of BAPTA loading should be confirmed, by monitoring calcium levels before and after BAPTA loading. Moreover, in addition to Chr2, an additional agonist should be used to test that. This is an important control that needs also to be shown.*

As we described in the items 1 and 2, we have confirmed with calcium imaging that astrocytic loading BAPTA blocked both light- and ATP-induced calcium elevation in astrocytes (Fig 2, see also Fig R1).

Figure R1. ATP-induced calcium signal in control (blue) and BAPTA-loaded (green) astrocytes.

6. *Authors assessed the effects of ChR2-stimulation of astrocytes analyzing the spontaneous firing rates of the recorded neurons. This parameter is subject to multiple variables (e.g., it is very sensitive to small changes in resting potential, internal solution, intracellular rundown mechanisms, etc). The reported experiments directly show changes in neuronal excitability, but standard procedures more accurate and well established in the literature needs to be used to support the validity of the conclusion. Specifically, recordings in voltage-clamp conditions monitoring changes in the membrane current and the membrane conductance.*

Authors have adequately done that when puffing ATP (e.g., Fig. 3A-C), but this should also be done in other experimental approaches and, more importantly, in ChR2 experiments.

We agree that monitoring changes in the membrane current and the membrane conductance are usually done in voltage clamp mode, but still there are many people have done these in current clamp mode (Borin et al., *Frontiers in cellular neuroscience* 2014; Gao and Derbenev, *J Neurophysiol* 2013; Gao and Smith, *J Neurophysiol* 2010; Gorelova et al., *J Neurophysiol* 2002) . Given that ChR2 activation induced increase of extracellular ATP is relatively low ($\sim 1 \mu\text{M}$, Fig. 1F), the light induced membrane current changes in voltage clamp mode is reluctant. In current clamp the light induced changes of membrane potential is still low, but small voltage change would induce robust action potential frequency changes. Thus the action potential firing serves as a signal amplifier of membrane potential changes. That is the main reason why we use current clamp rather than voltage clamp in most of our experiments. To keep consistency, we thus tested the input resistance in current clamp mode. However, to follow the reviewer's suggestion we monitored input resistance changes in ChR2 experiments and found that light stimuli indeed increased input resistance in interneurons and decreased input resistance in pyramidal neurons (Fig. S7), consistent with the ATP-induced changes in membrane input resistance (Fig. 5 C and D, Fig. 6 C and D).

7. *Moreover, ATP is known to elevate calcium in hippocampal astrocytes, which may lead to the release of other gliotransmitters, including ATP. This may affect the pharmacological analysis performed, because the sensitivity of the neuronal effects to purinergic antagonists may be due to indirect blockade of astrocytic receptors.*

We agree that results obtained from pharmacological studies are subject to complications by indirect effects. Our major conclusions are thus not based on only single result from pharmacological analysis. To avoid indirect effects of subsequent glutamate release, we applied ionotropic glutamate receptor antagonists in all experiments and also verified that metabolic glutamate receptor antagonists AP3

cannot block the ATP effect (Fig. 4 B and E). Furthermore, we found that P2Y1 and A1 purinergic antagonists MRS2179 and DPCPX, which largely blocked ATP-induced increase and decrease of the excitability of interneurons and pyramidal neurons respectively, only slightly inhibited the ATP induced calcium elevation in astrocyte (Fig R2). Thus ATP-induced calcium elevation in astrocytes (possibly through multiple purinergic receptors), which should be responsible for gliotransmitter release if any, is unlikely involved in the ATP-induced differential effects on neuronal excitabilities.

Figure R2. ATP-induced calcium signal in control (blue) and in the presence of 10 μ M MRS2179 (green) or 1 μ M DPCPX (red).

8. *ATP is known to regulate hippocampal synaptic transmission. At the beginning of the manuscript (Pg 5), it is stated that experiments were performed in the presence of ionotropic glutamate and GABA receptor antagonists. Authors should clarify whether this is the case for this initial experiments or apply to the rest of the manuscript (except when monitoring synaptic currents). I assume that is the case, otherwise it would complicate the possible interpretations by adding synaptic effects. Authors should clearly state the experimental conditions that apply.*

We appreciate the reviewer's comments. Indeed, we performed all the other experiments in the presence of ionotropic glutamate and GABA receptor antagonists. We added the following statement into the revised manuscript: "To exclude the influence of synaptic transmission, 0.5 mM kynurenic acid (an ionotropic glutamate receptor antagonist) and 10 μ M bicuculline (a GABA_A receptor antagonist) were applied to all experiments unless where specified" (page 5 line 8) .

9. *In Pg 6 authors state that "The onset of the responses of exogenous ATP is much faster than light stimulation, reflecting the feature of ATP exocytosis from astrocytes". As indicated above, since calcium levels were not analyzed, it remains unknown whether the different delays are due to the feature of ATP exocytosis or a delayed upstream mechanism such as delayed calcium elevations. Once again, the analysis of the astrocyte calcium responses is necessary.*

We agree that the delayed calcium elevation may also contribute to the different onsets of responses. Indeed, our calcium imaging results showed that the astrocyte calcium elevation delayed after light stimulation (Fig. 2A-C and G, supplemental movie 1). We have rephrased the statement in the revised manuscript (Page 6 line 13).

10. *In most of the pharmacological experiments, authors claim that the ATP-induced effect is blocked. However, the responses are significantly reduced but not completely abolished. These partial blockage suggest additional mechanisms in addition to the simple interpretation provided. How authors interpret such partial reductions?*

It is true that although ATP-induced excitability changes were largely blocked by specific A1 and P2Y1 receptor antagonists, some responses were only partially inhibited by other antagonists in some

experiments. It is known that effects of competitive antagonists depend on both binding efficiency and concentration. To avoid side effects of drugs, we only used reasonable concentration of antagonists which may explain the partial, rather than complete inhibition of some antagonists on the ATP-induced responses, although other possibilities cannot be completely excluded.

11. In Pg 22, the discussion of the mechanisms of both ChR2 stimulation and gliotransmitter release is too speculative and greatly superficial. Authors could contribute to clarify these issues by analyzing the astrocyte calcium evoked by ChR2 and its source.

We have rephrased these sentences and provide more evidence and references (page 20, 2nd paragraph).

Reviewer #2 (Remarks to the Author):

This manuscript describes the differential role of astrocytic ATP in CCK positive interneurons and CA1 pyramidal neurons in mouse hippocampus. Authors claim astrocytic ATP excites CCK positive interneurons via P2Y1 and by inhibiting K2P channels while ATP (after degradation by ectonuclease to adenosine) inhibits pyramidal neurons via A1 and activating GIRK channels.

Although the study is interesting and loaded with massive amount of data, there are several important issues to be resolved and clarified.

1. The use of GFAP-cre mouse is known to be problematic due to expression of GFAP in neuronal progenitor cells.

We indeed experienced such problem when using GFAP-Cre Floxp transgene mouse in our previous study (Wen et al., PLoS One 2013). In the present study we injected AAV virus in P18 when most if not all GFAP-positive cells in hippocampus are astrocyte as shown by immunostaining (Fig. S1). To further exclude the possibility we used inducible GFAP-Cre-ER line for all the new experiments performed for the revised manuscript. Virus was injected at P25 and Cre recombinase expression was induced at the same day. Experiments were performed 10 days after induction (>P35). We found that the light-induced neuronal excitability changes were reproduced in these mice (Fig. R3). These data has now been incorporated into Figure 1C.

Fig. R3. Light stimulation of astrocytes evoked the increase and decrease of the action potential frequency, respectively, in interneurons (left) and pyramidal neurons (right). ChR2 was specifically transfected into astrocytes by injection of AAV virus bearing ChR2 into hippocampus of GFAP-Cre-ER mice.

2. In Fig. S1D, authors need to count more than 15 or 19 cells (perhaps over 100 cells) to make sure that ChR2-mCherry is expressed only in astrocytes, but not in neurons or NG2 positive cells.

Actually the numbers (n) indicated in Fig S1D of the original manuscript were number of slices we examined rather than cell numbers. We are sorry for the confusion. In the revised manuscript we presented numbers in Fig S1D as total cell numbers counted, which were around 100.

3. The materials and method should contain at least minimal information regarding the virus and mouse information regarding how ChR2-mCherry virus was constructed, introduced into mouse brain, and where it was injected, instead of simply referring to Supplementary Info.

Following the reviewer's suggestion we described these details in the Materials and Method part of the revised manuscript (page 21).

4. Use of ChR2 for Ca²⁺ entry is known to be problematic due to its low Ca²⁺ permeability and undesirable changes of intracellular pH. (See, Optogenetic countering of glial acidosis suppresses glial glutamate release and ischemic brain damage.

Beppu K, Sasaki T, Tanaka KF, Yamanaka A, Fukazawa Y, Shigemoto R, Matsui K. Neuron. 2014 Jan 22;81(2):314-20.)

It is true that the permeability of ChR2 to Ca²⁺ is low. On the other hand, previous studies indeed demonstrated that ChR2 activation in astrocytes effectively increases intracellular Ca²⁺ (Chen et al., *Glia* 2013; Perea et al., *Nat Commun* 2014; Yang et al., *Cell Calcium* 2015), possibly through Ca²⁺/Na⁺ exchanger due to the high Na⁺ permeability of ChR2 channels (Yang et al., *Cell Calcium* 2015).

5. In page 5, the use of EGFP for control light stimulation needs to be described in the materials and method or results section.

We described that in the materials and method (page 21 line 14).

6. Repetitive depolarization of patched astrocyte in order to spread the BAPTA quickly is nice but this method can result in undesirable changes in physiology of astrocytes.

We agree that repetitive depolarization of patched astrocyte may affect physiology of astrocyte. We thus only injected very limited current (5 mV depolarization, 50 ms duration, 10Hz) to minimize the potential side effects. We also confirmed that the spontaneous calcium waves seemed to be normal when we loaded Alexa-647 alone into astrocyte through current injection (Fig R4), suggesting that the physiology of the astrocytes was largely normal in our experimental conditions.

Fig. R4. Loading Alexa-647 alone does not affect astrocyte spontaneous calcium signals. (A) Calcium signals of astrocytes bulk loaded with fluo-4 and intracellularly loaded with Alexa-647. (B) Calcium

signals of astrocytes bulk loaded with fluo-4. These two examples were collected from slices of the same mouse.

7. In page 6, authors need to qualify the statement, "The onset of the responses of exogenous ATP is much faster than light stimulation" by measuring the rise time and comparing the values.

We followed the reviewer's suggestion and quantified the time to peak parameters. The results were shown in figure 1J.

8. In page 7, authors need to quantify the frequency and amplitude of sPSPs.

We followed the reviewer's suggestion and quantified the frequency and amplitude of sPSPs. The results were integrated in Figure S4 E and F.

9. In Fig. 2A, the depolarizing current-induced AP's and firing pattern and ATP response in the same cell should be displayed. Each example of regularly spiking- and fast spiking-interneuron should be displayed.

We followed the reviewer's suggestion and added sample traces of regularly spiking and fast spiking interneurons in figure 3A. We also added some sample traces to show the firing pattern of CR, CB, NPY, and VIP in figure S6.

10. In page 12, lidocaine and quinidine are known to affect AP, especially lidocaine. However, Fig. 4K (Fig. 4E as well) shows no sign of inhibition of AP firing in the presence of lidocane or quinidine. How is this possible?

We thank the reviewer for pointing that out. Indeed, lidocaine and quinidine block action potential at high concentrations (more than 1 mM) (Gold et al., *J Pharmacol Exp Ther* 1998; Lang et al., *Anesthesiology* 2007; Scholz et al., *J Neurophysiol* 1998). In our experiments, we only perfused slices with relatively low concentration (100 μ M) of quinidine or lidocaine to examine the light-induced increase of action potential firing (Fig 5K and 5L). In such experiments, 5-20 pA more currents was injected into the recorded neurons to overcome the drug's inhibition on the threshold of action potential firing so that the effect of light stimulation could be tested (Fig R5 a and c). We only used higher concentration (1 mM) lidocaine to test ATP-induced depolarization (rather than action potential firing) of interneurons. We have addressed the issue in the revised manuscript (page 12, 2nd paragraph).

Fig. R5. Inhibition of neuronal firing by low concentration of lidocaine or quinidine can be overcome by intracellular injection of a little bit more current.

11. In page 12, the evidence TASK-3 is very weak and circumstantial. Authors should either tone down the TASK3 in abstract or directly demonstrate by using shRNA or KO mouse.

Following the reviewer's suggestion, we define the channel as K2P type channels rather than specifically TASK-3 subtype. Changes have been made in the abstract, the results part (page 13, 1st paragraph) and the discussion (page 18).

12. At first, Fig. 5A appears to be an I-V relationship, but it is not. Because readers are used to looking at I-V relationship, Fig. 5A is very confusing. For example, Fig. 4B is a typical I-V relationship. It is difficult to understand why authors decided to show Fig. 5A not as an I-V relationship.

We apologize for the confusing. In the revised manuscript we have characterized the I-V relationship of ATP-induced current in pyramidal neurons using the same protocol as used for interneurons (Fig. 6 A).

13. In page 14, bumetanide is misspelled as "bumitanide."

We thank the reviewer for pointing that out. Correction has been made.

14. In page 16, it is interesting to see the endogenous tonic level of ATP in brain slices. However, it is difficult to conclude that it is coming from astrocytes without further evidence. Therefore, I suggest that authors should either take out this part (Fig. 7) from this study to save for future paper or perform additional experiments to demonstrate that it is coming from astrocytes.

Following the reviewer's suggestion, we have taken out this result in the revised manuscript.

15. In Fig. 8, it is interesting to see the possible involvement of astrocytic ATP in gamma oscillation. However, I realized that the experiment was done on acute brain slices, but not in whole intact animal. It will be difficult to say that gamma oscillation observed in brain slices would be the same gamma oscillation in intact animals. Therefore, I suggest to call this gamma oscillation with different name, such as "gamma oscillation in brain slice" or "ex vivo gamma oscillation" to distinguish it from real gamma oscillation that we observe in intact animals. This word needs to be replaced in entire manuscript.

We changed our statements to "ex vivo gamma oscillation" as the reviewer suggested.

16. The discussion is too long and verbose. It needs to be shortened. Especially the section on interneuron subtypes and characterization needs to be shortened.

The discussion part was shortened and polished.

Reviewer #3 (Remarks to the Author):

In this manuscript, the authors have investigated the regulation of K channels by ATP released from astrocytes. There have been previous reports suggesting extracellular ATP regulated K⁺ channels, but none of these studies were performed under normal receptor and channel expression levels in brain slices. The general interest in this paper would be high, given the novel discovery of another mechanism by

which astrocytes regulate neuronal excitability. The basic strategy that the authors used to approach this question in vivo was to conditionally (GFAP-Cre) express a light-activated channelrhodopsin in astrocytes. They found increased excitability of CCK interneurons (via K2P by P2Y1Rs) and decreased excitability of pyramidal neurons due to opening of GIRK via activation of A1 receptors.

The manuscript is reasonably well-written, although it could use another pass of editing to make the dense text easier to read. The data presented in the figures are impressive, convincing and the primary conclusions presented by the authors are supported. The Authors do rely heavily on pharmacology for their primary conclusions, but did utilize the P2Y1R KO mouse to strengthen their findings. The potential involvement of the newly defined two-pore domain K channel is very interesting, specifically the TASK family in the hippocampus. Evidence for this statement, again, is primarily based on pharmacological profiles, but their data are consistent and supportive. Impressive immunocytochemistry data are also supportive of the necessary receptors and channels being in the appropriate cell.

ATP clearly induces a hyperpolarization and its K⁺ channel based on reversal potentials. GIRK channel blockers point to its regulation being directly mediated by the betagamma subunit (as well as PTX). This same pharmacological profile was repeated when ATP was specifically released from astrocyte activation.

As noted above, the main text is very dense reading because of all the pharmacology. A final diagram/cartoon of all the major players (cells, channels and receptors) would be very welcome and make the arguments in the manuscript much easier to follow.

We thank the reviewer for the positive comments and valuable suggestions. We added a model cartoon as figure 9.

A few minor issues:

Figure legend of S1, D and E are not correct, D-F in figure misidentified. Please reference the statement "passive membrane properties expected of astrocytes".

We thank the reviewer for pointing out our mistake. We have corrected these mistakes and added a reference for the statement (Bergles and Jahr, Neuron 1997).

In the text, p7, line 4, the authors state "The decreased sPSPs may reflect the direct inhibitory effect of ATP on synaptic transmission through presynaptic A1 receptors". Authors need to reference this statement.

We added the reference (Wu and Saggau, Neuron 1994) .

p. 11, their statement, based on pharmacology only, "M-current inhibition is not involved in the ATP-evoked increase in interneuron excitability" is too strong, please soften statement.

We changed to "M-current inhibition may not be involved in the ATP-evoked increase in interneuron excitability"

Reply to reviewers' comments reference:

Bergles, D.E., and Jahr, C.E. (1997). Synaptic activation of glutamate transporters in hippocampal astrocytes. *Neuron* 19, 1297-1308.

Borin, M., Fogli Iseppe, A., Pignatelli, A., and Belluzzi, O. (2014). Inward rectifier potassium (Kir) current in dopaminergic periglomerular neurons of the mouse olfactory bulb. *Frontiers in cellular neuroscience* 8, 223.

Chen, J., Tan, Z., Zeng, L., Zhang, X., He, Y., Gao, W., Wu, X., Li, Y., Bu, B., Wang, W., *et al.* (2013). Heterosynaptic long-term depression mediated by ATP released from astrocytes. *Glia* *61*, 178-191.

Gao, H., and Derbenev, A.V. (2013). Synaptic and extrasynaptic transmission of kidney-related neurons in the rostral ventrolateral medulla. *J Neurophysiol* *110*, 2637-2647.

Gao, H., and Smith, B.N. (2010). Tonic GABAA receptor-mediated inhibition in the rat dorsal motor nucleus of the vagus. *J Neurophysiol* *103*, 904-914.

Gorelova, N., Seamans, J.K., and Yang, C.R. (2002). Mechanisms of dopamine activation of fast-spiking interneurons that exert inhibition in rat prefrontal cortex. *J Neurophysiol* *88*, 3150-3166.

Gold, M.S., Reichling, D.B., Hampl, K.F., Drasner, K., and Levine, J.D. (1998). Lidocaine toxicity in primary afferent neurons from the rat. *J Pharmacol Exp Ther* *285*, 413-421.

Lang, P.M., Hilmer, V.B., and Grafe, P. (2007). Differential contribution of sodium channel subtypes to action potential generation in unmyelinated human C-type nerve fibers. *Anesthesiology* *107*, 495-501.

Perea, G., Yang, A., Boyden, E.S., and Sur, M. (2014). Optogenetic astrocyte activation modulates response selectivity of visual cortex neurons in vivo. *Nat Commun* *5*, 3262.

Scholz, A., Kuboyama, N., Hempelmann, G., and Vogel, W. (1998). Complex blockade of TTX-resistant Na⁺ currents by lidocaine and bupivacaine reduce firing frequency in DRG neurons. *J Neurophysiol* *79*, 1746-1754.

Wen, J., Yang, H.B., Zhou, B., Lou, H.F., and Duan, S. (2013). beta-Catenin is critical for cerebellar foliation and lamination. *PLoS One* *8*, e64451.

Wu, L.G., and Saggau, P. (1994). Adenosine inhibits evoked synaptic transmission primarily by reducing presynaptic calcium influx in area CA1 of hippocampus. *Neuron* *12*, 1139-1148.

Yang, J., Yu, H., Zhou, D., Zhu, K., Lou, H., Duan, S., and Wang, H. (2015). Na⁽⁺⁾-Ca⁽²⁾(⁺) exchanger mediates ChR2-induced [Ca⁽²⁾(⁺)]_i elevation in astrocytes. *Cell Calcium* *58*, 307-316.

REVIEWERS' COMMENTS:

Reviewer #1 (Remarks to the Author):

Authors have adequately and reasonably addressed all the concerns about the previous version. The interesting conclusions reached are now more solidly supported by the experimental results.

I have no further comments.

Reviewer #2 (Remarks to the Author):

Authors have addressed all the comments satisfactorily.

Reviewer #3 (Remarks to the Author):

The authors have addressed my previous concerns.

REVIEWERS' COMMENTS:

Reviewer #1 (Remarks to the Author):

Authors have adequately and reasonably addressed all the concerns about the previous version. The interesting conclusions reached are now more solidly supported by the experimental results.

I have no further comments.

We thank the reviewer for the positive comments.

Reviewer #2 (Remarks to the Author):

Authors have addressed all the comments satisfactorily.

We thank the reviewer for the positive comments.

Reviewer #3 (Remarks to the Author):

The authors have addressed my previous concerns.

We thank the reviewer for the positive comments.